# Enhanced Federated Optimization:
# Adaptive Unbiased Sampling with Reduced Variance

## Abstract

Federated Learning (FL) is a distributed learning paradigm to train a global model across multiple devices without collecting local data. In FL, a server typically selects a subset of clients for each training round to optimize resource usage. Central to this process is the technique of unbiased client sampling, which ensures a representative selection of clients. Current methods primarily utilize a random sampling procedure which, despite its effectiveness, achieves suboptimal efficiency owing to the loose upper bound caused by the sampling variance. In this work, by adopting an independent sampling procedure, we propose a federated optimization framework focused on adaptive unbiased client sampling, improving the convergence rate via an online variance reduction strategy. In particular, we present the first adaptive client sampler, K-Vib, employing an independent sampling procedure. K-Vib achieves a linear speed-up on the regret bound $\tilde{\mathcal{O}}\big(N^{\frac{1}{3}}T^{\frac{2}{3}}/K^{\frac{4}{3}}\big)$ within a set communication budget $K$. Empirical studies indicate that K-Vib doubles the speed compared to baseline algorithms, demonstrating significant potential in federated optimization.

## 1 Introduction

This paper studies the prevalent cross-device federated learning (FL) framework, as outlined in Kairouz et al. (2021), which optimizes $\boldsymbol{x} \in \mathcal{X} \subseteq \mathbb{R}^d$ to minimize a finite-sum objective:

$$\min_{\boldsymbol{x} \in \mathcal{X}} f(\boldsymbol{x}) := \sum_{i=1}^{N} \boldsymbol{\lambda}_i f_i(\boldsymbol{x}) := \sum_{i=1}^{N} \boldsymbol{\lambda}_i \mathbb{E}_{\xi_i \sim \mathcal{D}_i}[F_i(\boldsymbol{x}, \xi_i)], \tag{1}$$

where $N$ denotes the total number of clients, and $\boldsymbol{\lambda}$ denotes the weights of client objective ($\boldsymbol{\lambda}_i \geq 0, \sum_{i=1}^{N} \boldsymbol{\lambda}_i = 1$). The local loss function $f_i : \mathbb{R}^d \to \mathbb{R}$ is intricately linked to the local data distribution $D_i$. It is defined as $f_i(\boldsymbol{x}) = \mathbb{E}_{\xi_i \sim \mathcal{D}_i}[F_i(\boldsymbol{x}, \xi_i)]$, where $\xi_i$ represents a stochastic batch drawn from $D_i$. Federated optimization algorithms, such as FEDAVG (McMahan et al., 2017), are designed to minimize objectives like equation 1 by alternating between local and global updates in a distributed learning framework. To reduce communication and computational demands in FL (Konečný et al., 2016; Wang et al., 2021; Yang et al., 2022), various client sampling strategies have been developed (Chen et al., 2020; Cho et al., 2020b; Balakrishnan et al., 2022; Wang et al., 2023; Malinovsky et al., 2023; Cho et al., 2023). These strategies are crucial as they decrease the significant variations in data quality and volume across clients (Khan et al., 2021). Thus, efficient client sampling is key to enhancing the performance of federated optimization.

Current sampling methodologies in FL are broadly divided into biased (Cho et al., 2020b; Balakrishnan et al., 2022; Chen & Vikalo, 2023) and unbiased categories (El Hanchi & Stephens, 2020; Wang et al., 2023). Unbiased client sampling holds particular significance as it maintains the consistency of the optimization objective Wang et al. (2023; 2020). Specifically, unlike biased sampling where client weights $\boldsymbol{\lambda}$ are proportional to sampling probabilities, unbiased methods separate these weights from sampling probabilities. This distinction enables unbiased sampling to be integrated effectively with strategies that address data heterogeneity (Zeng et al., 2023c), promote fairness (Li et al., 2020c;a), and enhance robustness (Li et al., 2021; 2020a). Additionally, unbiased sampling aligns with secure aggregation protocols for confidentiality in FL (Du & Atallah, 2001;

---

**Algorithm 1** FedAvg with Unbiased Client Sampler

---

**Require:** Client set $S$, where $|S| = N$, client weights $\boldsymbol{\lambda}$, times $T$, local steps $R$
1: Initialize sample distribution $\boldsymbol{p}^0$ and model $\boldsymbol{x}^0$
2: **for** time $t$ in $[T]$ **do**
3:     Server runs sampling procedure to create $S^t \sim \boldsymbol{p}^t$
4:     Server broadcasts $\boldsymbol{x}^t$ to sampled clients $i \in S^t$
5:     **for** each client $i \in S^t$ in parallel **do**
6:         $\boldsymbol{x}_i^{t,0} = \boldsymbol{x}^t$
7:         **for** local steps $r$ in $[R]$ **do**
8:             $\boldsymbol{x}_i^{t,r} = \boldsymbol{x}_i^{t,r-1} - \eta_l \nabla F_i(\boldsymbol{x}_i^{t,r-1})$
9:         **end for**
10:        Client uploads local updates $\boldsymbol{g}_i^t = \boldsymbol{x}_i^{t,0} - \boldsymbol{x}_i^{t,R}$
11:     **end for**
12:     Server builds estimates $\boldsymbol{d}^t = \sum_{i \in S^t} \boldsymbol{\lambda}_i \boldsymbol{g}_i^t / \boldsymbol{p}_i^t$
13:     Server updates $\boldsymbol{x}^{t+1} = \boldsymbol{x}^t - \eta_g \boldsymbol{d}^t$
14:     Server updates $\boldsymbol{p}^{t+1}$ based on $\{\|\boldsymbol{g}_i^t\|\}_{i \in S^t}$
15: **end for**

---

Goryczka & Xiong, 2015; Bonawitz et al., 2017). Hence, unbiased client sampling techniques are indispensable for optimizing federated systems.

Therefore, a better understanding of the implications of unbiased sampling in FL could help us to design better algorithms. To this end, we summarize a general form of federated optimization algorithms with unbiased client sampling in Algorithm 1. Despite differences in methodology, the algorithm covers unbiased sampling techniques (Wang et al., 2023; Malinovsky et al., 2023; Cho et al., 2023; Salehi et al., 2017; Borsos et al., 2018; El Hanchi & Stephens, 2020; Zhao et al., 2021b) in the literature. In Algorithm 1, unbiased sampling comprises three primary steps (referring to lines 3, 12, and 14). First, the **Sampling Procedure** generates a set of samples $S^t$ along with their respective probabilities. Second, the **Global Estimation** step creates global estimates for model updates, aiming to approximate the outcomes as if all participants were involved. Finally, the **Adaptive Strategy** adjusts the sampling probabilities based on the incoming information, ensuring dynamic adaptation to changing data conditions.

Typically, unbiased sampling methods in FL are founded on a **random sampling procedure**, which is then refined to improve global estimation and adaptive strategies. However, the exploration of alternative sampling procedures to enhance unbiased sampling has not been thoroughly investigated. Our research shifts focus to the **independent sampling procedure**, a less conventional approach yet viable for FL. We aim to delineate the distinctions between these methodologies as follows.

> *Random sampling procedure (RSP) means that the server samples clients from a black box without replacement.*

> *Independent sampling procedure (ISP) means that the server rolls a dice for every client independently to decide whether to include the client.*

Building on the concept of arbitrary sampling (Horváth & Richtárik, 2019; Chen et al., 2020), our study observes that the independent sampling procedure can enhance the efficiency of estimating full participation outcomes in FL servers, as detailed in Section 3. However, integrating independent sampling into unbiased techniques introduces new constraints, as outlined in Remark 2.1. Addressing this innovatively in Lemma 5.1, our paper studies the effectiveness of general FL algorithms with adaptive unbiased client sampling, particularly emphasizing the utility and implications of the independent sampling procedure from an optimization standpoint.

**Our Contributions**: This paper presents a comprehensive analysis of the non-convex convergence in FedAvg and its variants. We first establish a novel link between the cumulative variance of global estimates and

convergence rates by separating global estimation results from heterogeneity-related factors. Thus to reduce the cumulative variance, we introduce K-VIB, a novel adaptive sampler incorporating the independent sampling procedure. K-VIB notably achieves an expected regret bound of $\tilde{\mathcal{O}}\big(N^{\frac{1}{3}}T^{\frac{2}{3}}/K^{\frac{4}{3}}\big)$, demonstrating a near-linear speed-up over existing bounds $\tilde{\mathcal{O}}\big(N^{\frac{1}{3}}T^{\frac{2}{3}}\big)$ (Borsos et al., 2018) and $\mathcal{O}\big(N^{\frac{1}{3}}T^{\frac{2}{3}}\big)$ (El Hanchi & Stephens, 2020). Empirically, K-VIB shows accelerated convergence on standard federated tasks compared to baseline algorithms.

## 2 Preliminaries

We first introduce previous works on batch sampling (Horváth & Richtárik, 2019) in stochastic optimization and optimal client sampling (Chen et al., 2020) in FL. We made a few modifications to fit our problem setup.

**Remark 2.1.** *We define communication budget $K$ as the expected number of sampled clients. And, its value range is from $1$ to $N$. To be consistent, the sampling probability $\boldsymbol{p}$ always satisfies the constraint $\boldsymbol{p}_i^t > 0, \sum_{i=1}^N \boldsymbol{p}_i^t = K, \forall t \in [T]$ in this paper.*

**Definition 2.1** (Unbiasedness of client sampling $S^t$). *For communication round $t \in [T]$, the estimator $\boldsymbol{d}^t$ is related to sampling probability $\boldsymbol{p}^t$ and the sampling procedure $S^t \sim \boldsymbol{p}^t$. We define a client sampling as unbiased if the sampling $S^t$ and estimates $\boldsymbol{d}^t$ satisfy that*

$$\mathbb{E}[\boldsymbol{d}^t] = \mathbb{E}[\sum_{i \in S^t} \boldsymbol{\lambda}_i \boldsymbol{g}_i^t / \boldsymbol{p}_i^t] = \sum_{i=1}^N \boldsymbol{\lambda}_i \boldsymbol{g}_i^t.$$

*Besides, the variance of estimator $\boldsymbol{d}^t$ can be derived as:*

$$\mathbb{V}(S^t) := \mathbb{E}_{S^t \sim \boldsymbol{p}^t}\left[\left\|\sum_{i \in S^t} \frac{\boldsymbol{\lambda}_i \boldsymbol{g}_i^t}{\boldsymbol{p}_i^t} - \sum_{i=1}^N \boldsymbol{\lambda}_i \boldsymbol{g}_i^t\right\|^2\right], \tag{2}$$

*where $\mathbb{E}[|S^t|] = K$. We omit the terms $\boldsymbol{\lambda}, \boldsymbol{g}^t$ for notational brevity.*

**Optimal unbiased client sampling.** Optimal unbiased client sampling should achieve the lowest variance, i.e., equation 2. It is to estimate the global gradient of full-client participation, *i.e.,*minimize the variance of estimator $\boldsymbol{d}^t$. Given a fixed communication budget $K$, the optimality of the global estimator depends on the collaboration of sampling distribution $\boldsymbol{p}^t$ and the corresponding procedure that outputs $S^t$.

In detail, different sampling procedures associated with the sampling distribution $\boldsymbol{p}$ build a different *probability matrix* $\mathbf{P} \in \mathbb{R}^{N \times N}$, with the elements defined as $\mathbf{P}_{ij} := \mathrm{Prob}(\{i, j\} \subseteq S)$. Arbitrary sampling (Horváth & Richtárik, 2019) has shown the generality of denoting arbitrary sampling procedure with a probability matrix for stochastic optimization. Inspired by their findings, we focus on the optimal sampling procedure for the FL server in Lemma 2.1.

**Lemma 2.1** (Optimal sampling procedure, Horváth & Richtárik, 2019). *For any communication round $t \in [T]$ in FL, random sampling yielding the $\mathbf{P}_{ij}^t = Prob(i, j \in S^t) = K(K-1)/N(N-1)$, and independent sampling yielding $\mathbf{P}_{ij}^t = Prob(i, j \in S^t) = \boldsymbol{p}_i^t \boldsymbol{p}_j^t$, they admit*

$$\mathbb{V}(S^t) = \underbrace{\sum_{i=1}^N (1 - \boldsymbol{p}_i^t) \frac{\boldsymbol{\lambda}_i^2 \|\boldsymbol{g}_i^t\|^2}{\boldsymbol{p}_i^t}}_{independent\ sampling} \leq \underbrace{\frac{N-K}{N-1} \sum_{i=1}^N \frac{\boldsymbol{\lambda}_i^2 \|\boldsymbol{g}_i^t\|^2}{\boldsymbol{p}_i^t}}_{random\ sampling}. \tag{3}$$

The lemma indicates that the independent sampling procedure is the optimal sampling procedure that minimizes the upper bound of variance. Then, we have the optimal probability by solving the minimization of the upper bound in respecting probability $\boldsymbol{p}$ in Lemma 2.2.

**Lemma 2.2** (Optimal sampling probability, Chen et al., 2020). *Generally, we can let $\boldsymbol{a}_i = \boldsymbol{\lambda}_i \|\boldsymbol{g}_i^t\|, \forall i \in [N], t \in [T]$ for simplicty of notation. Assuming $0 < \boldsymbol{a}_1 \leq \boldsymbol{a}_2 \leq \cdots \leq \boldsymbol{a}_N$ and $0 < K \leq N$, and $l$ is the largest*

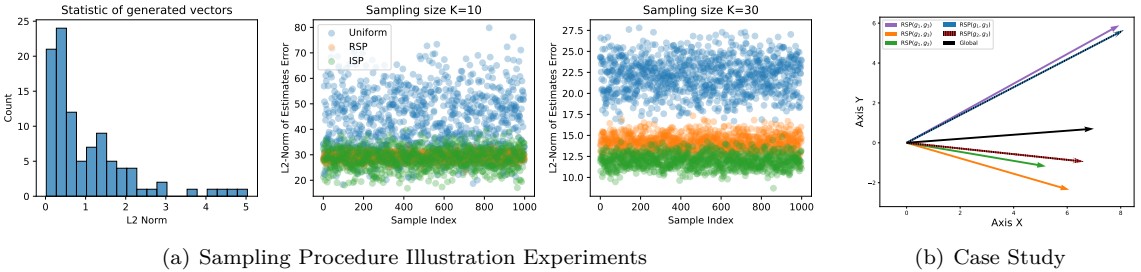

(a) Sampling Procedure Illustration Experiments    (b) Case Study

Figure 1: The variance of ISP estimates is lower than RSP. Global estimates on the X-Y plane. (a) Scatter plot of estimates errors, where "uniform" indicates the RSP with uniform probability. (b) The notations $\mathrm{RSP}(\boldsymbol{g}_i, \boldsymbol{g}_j)$ and $\mathrm{ISP}(\boldsymbol{g}_i, \boldsymbol{g}_j)$ represent the global estimates constructed through random sampling and independent sampling, respectively, using sampled vectors $\boldsymbol{g}_i$ and $\boldsymbol{g}_j$. Global indicates the full participation results. We can see $\mathrm{ISP}(\boldsymbol{g}_i, \boldsymbol{g}_j)$ is closer to Global.

*integer for which $0 < K + l - N \le \frac{\sum_{i=1}^{l} \boldsymbol{a}_i}{\boldsymbol{a}_l}$, we have*

$$\boldsymbol{p}_i^* = \begin{cases} (K + l - N)\dfrac{\boldsymbol{a}_i}{\sum_{j=1}^{l} \boldsymbol{a}_j}, & \text{if } i \le l, \\ 1, & \text{if } i > l, \end{cases} \qquad (ISP) \qquad (4)$$

*to be a solution to the optimization problem $\min_{\boldsymbol{p}} \sum_{i=1}^{N} \frac{\boldsymbol{a}_i^2}{\boldsymbol{p}_i}$. In contrast, we provide the optimal sampling probability for the random sampling procedure*

$$\boldsymbol{p}_i^* = K \frac{\boldsymbol{a}_i}{\sum_{j=1}^{N} \boldsymbol{a}_j}. \qquad (RSP) \qquad (5)$$

Therefore, the optimal client sampling in FL uses ISP with probability given in equation 4.

## 3 Case Study on Sampling Procedure

We suggest designing sampling probability for the ISP to enhance the power of unbiased client sampling in federated optimization. Beyond the tighter upper bound of variance in equation 3, three main properties can demonstrate the superiority of ISP over RSP. We illustrate the points via Example 3.1 and Example 3.2.

**Example 3.1.** *We randomly generate 100 vectors with the size of 1000 dimensions. Then, we run 1000 times the RSP and ISP with its best probability in Lemma 2.2 to estimate full aggregation results. We draw the error of these estimate results, as shown in Figure 1(a). We note the mean of these points related to equation 2.*

**Example 3.2.** *We consider a case example $N = 3, K = 2$ with $\boldsymbol{g}_1 = (\frac{\sqrt{2}}{2}, \frac{\sqrt{2}}{2}), \boldsymbol{g}_2 = (1, -2\sqrt{2}), g_3 = (2\sqrt{7}, 2\sqrt{2})$, inducing weights vector $[\|\boldsymbol{g}_1\|^2, \|\boldsymbol{g}_2\|^2, \|\boldsymbol{g}_3\|^2] = [1, 3, 6]$ if we omit $\boldsymbol{\lambda}$. We have optimal sampling probability $\boldsymbol{p}^* = K \cdot [0.1, 0.3, 0.6]$ for random sampling procedure and $\boldsymbol{p}^* = [0.25, 0.75, 1]$ for independent sampling procedure. We depict the possible sampling results in Figure 1(b).*

**Random sampling probability is a special case of independent sampling.** With a minimum budget of $K = 1$, the ISP does not assign any client with probability 1, it returns to the random sampling solution according to equation 4. If the budget $K > 1$, the solution of the independent sampling procedure will change, while the RSP does not. Hence, it builds better estimates than random sampling with the same sampling results as shown in the example. This is because the optimal probability of random sampling is minimizing a loose upper bound of variance equation 3. The results tend to let each of the single estimates $\boldsymbol{a}_i/\boldsymbol{p}_i = \sum \boldsymbol{a}_i$.

**Independent sampling estimates are asymptotic to full participation results.** ISP builds estimates asymptotically to the full-participation results with an increasing communication budget of $K$, while random

sampling does not. As shown in Figure 1(a), RSP and ISP achieve comparable estimates errors with lower budget $K = 10$. Then, the ISP outperforms RSP with a larger budget of $K = 30$. Refer to Example 3.2, random sampling with full participation ($K = 3$) builds estimates $\boldsymbol{d}^t = (6.4, 5.9)$ and hence $\|\boldsymbol{d}^t - \sum \boldsymbol{g}_i\|^2 = 0.6$. Analogously, full participation induces $\boldsymbol{p}^* = (1, 1, 1)$ for independent sampling and hence $\|\boldsymbol{d}^t - \sum \boldsymbol{g}_i\|^2 = 0$.

**Independent sampling creates expected sampling size.** The number of sampling results from independent sampling is stochastic with expectation $K$. It means that if we strictly conduct the independent sampling procedure, the number of sampling results $\text{Prob}(|S^t| = K) \neq 1$, but $\mathbb{E}[|S^t|] = K$. Referring to the example, independent sampling may sample 3 clients with probability $p = 0.25 * 0.75 * 1 = 3/16$ and sample only 1 client with $p = (1 - 0.25) * (1 - 0.75) = 3/16$. Importantly, the perturbation of sampling results is acceptable due to the straggler clients (Gu et al., 2021) in a large-scale cross-device FL system. Besides, we can easily extend our analyses to the case with straggler as discussed in Appendix E.1.

Observing the superiority of ISP, we propose to design a sampling probability and global estimates with ISP in federated learning. However, computing the optimal sampling via equation 4 requires a norm of full gradients, which is unfeasible in practice. Therefore, FL needs a better design of its sampling probability for ISP based on limited information, that is unexplored. **Unless otherwise stated, all sampling probability $\boldsymbol{p}$ and sampling procedures are related to ISP in the remainder of this paper.**

## 4 General Convergence Analyses of FL with Unbiased Client Sampling

In this section, we first provide a general convergence analysis of FedAvg covered by Algorithm 1, specifically focusing on the variance of the global estimator. Our analysis aims to identify the impacts of sampling techniques on enhancing federated optimization. To this end, we define important concepts below to clarify the improvement given by an applied unbiased sampling:

**Definition 4.1** (Sampling quality). *Given communication budget $K$ and arbitrary unbiased client sampling probability $\boldsymbol{p}^t$, we measure the quality (lower is better) of one sampling step $S^t \sim \boldsymbol{p}^t$ by its expectation discrepancy to the optimal sampling:*

$$Q(S^t) := \mathbb{E}_{S^t \sim \boldsymbol{p}^t} \left[ \left\| \sum_{i \in S^t} \frac{\boldsymbol{\lambda}_i \boldsymbol{g}_i^t}{\boldsymbol{p}_i^t} - \sum_{i \in S_*^t} \frac{\boldsymbol{\lambda}_i \boldsymbol{g}_i^t}{\boldsymbol{p}_i^*} \right\|^2 \right], \tag{6}$$

*where $S_*^t \sim \boldsymbol{p}^*$ is the ISP, $\boldsymbol{p}^*$ is obtained via equation 4 with full $\{\|\boldsymbol{g}_i^t\|\}_{i \in [N]}$, and $\mathbb{E}[|S^t|] = \mathbb{E}[|S_*^t|] = K$.*

**Remark.** Note that the second term of equation 6 denotes the best results that can be possibly obtained subjected to communication budget. It still preserves estimate errors to full results. Therefore, we define the sampling quality of one sampling by its gap to the optimal estimate results for practical concern.

**Definition 4.2** (The optimal factor). *Given an iteration sequence of global model $\{\boldsymbol{x}^1, \ldots, \boldsymbol{x}^T\}$, under the constraints of communication budget $K$ and local updates statues $\{\boldsymbol{g}_i^t\}_{i \in [N]}, t \in [T]$, we define the improvement factor of applying optimal client sampling $S_*^t \sim \boldsymbol{p}^*$ comparing uniform sampling $U^t$ as:*

$$\alpha_*^t := \frac{\mathbb{E}\left[ \left\| \sum_{i \in S_*^t} \frac{\boldsymbol{\lambda}_i}{\boldsymbol{p}_i^*} \boldsymbol{g}_i^t - \sum_{i=1}^{N} \boldsymbol{\lambda}_i \boldsymbol{g}_i^t \right\|^2 \right]}{\mathbb{E}\left[ \left\| \sum_{i \in U^t} \frac{\boldsymbol{\lambda}_i}{\boldsymbol{p}_i} \boldsymbol{g}_i^t - \sum_{i=1}^{N} \boldsymbol{\lambda}_i \boldsymbol{g}_i^t \right\|^2 \right]},$$

*and optimal $\boldsymbol{p}^*$ is computed via equation 4 with $\{\boldsymbol{g}_i^t\}_{i \in [N]}$.*

**Remark.** The factor $\alpha_*^t \in [0, 1]$ denotes the optimal efficiency that one sampling technique can achieve under the current constraints $K, \{\boldsymbol{g}_i^t\}_{i \in [N]}, \boldsymbol{\lambda}$. Theoretically, $\alpha_*^t = 0$ denotes the best improvement obtained by minimizing equation 6. However, the optimal client sampling $Q(S^t) = 0$ can not be achieved without revealing, *i.e.,* communicating all clients' full updates to the server.

In practical settings, federated learning typically trains modern neural networks, which are non-convex problems in optimization. Therefore, our convergence analyses rely on standard assumptions on the local

empirical function $f_i, i \in [N]$ in non-convex federated optimization (Chen et al., 2020; Jhunjhunwala et al., 2022; Chen & Vikalo, 2023).

**Assumption 4.1** (Smoothness). *Each objective $f_i(\boldsymbol{x})$ for all $i \in [N]$ is L-smooth, inducing that for all $\forall \boldsymbol{x}, \boldsymbol{y} \in \mathbb{R}^d$, it holds $\|\nabla f_i(\boldsymbol{x}) - \nabla f_i(\boldsymbol{y})\| \leq L\|\boldsymbol{x} - \boldsymbol{y}\|$.*

**Assumption 4.2** (Unbiasedness and bounded local variance). *For each $i \in [N]$ and $\boldsymbol{x} \in \mathbb{R}^d$, we assume the access to an unbiased stochastic gradient $\nabla F_i(\boldsymbol{x}, \xi_i)$ of client's true gradient $\nabla f_i(\boldsymbol{x})$, i.e., $\mathbb{E}_{\xi_i \sim \mathcal{D}_i} [\nabla F_i(\boldsymbol{x}, \xi_i)] = \nabla f_i(\boldsymbol{x})$. The function $f_i$ have $\sigma_l$-bounded (local) variance i.e., $\mathbb{E}_{\xi_i \sim \mathcal{D}_i} \left[ \|\nabla F_i(\boldsymbol{x}, \xi_i) - \nabla f_i(\boldsymbol{x})\|^2 \right] \leq \sigma_l^2$.*

**Assumption 4.3** (Bounded global variance). *We assume the weight-averaged global variance is bounded, i.e., $\sum_{i=1}^{N} \boldsymbol{\lambda}_i \|\nabla f_i(\boldsymbol{x}) - \nabla f(\boldsymbol{x})\|^2 \leq \sigma_g^2$ for all $\boldsymbol{x} \in \mathbb{R}^d$.*

Assumptions 4.1 and 4.2 are standard assumptions in stochastic optimization analyses. Assumption 4.3 measures the impacts of data heterogeneity in federated optimization. A larger upper bound of $\sigma_g^2$ denotes stronger heterogeneity across clients. Now, we provide the non-convex convergence of Algorithm 1.

**Theorem 4.1** (FedAvg with decomposed unbiased sampling quality). *Under Assumptions 4.1, 4.2, 4.3, given an iteration sequence $\{\boldsymbol{x}^1, \ldots, \boldsymbol{x}^t\}$ generated by Algorithm 1, taking upper bound $\mathbb{E}\left[f(\boldsymbol{x}^1) - f(\boldsymbol{x}^*)\right] \leq M$, $\eta_g \leq \min\left(\sqrt{\frac{2M}{T\bar{\beta}}}, \frac{1}{L}\right)$, we have*

$$\min_{t \in [T]} \mathbb{E}\|\nabla f(\boldsymbol{x}^t)\|^2 \leq \sqrt{\frac{8M\bar{\beta} + T\epsilon^2}{T\hat{\rho}^2}} + \frac{\frac{1}{T}\sum_{t=1}^{T} Q(S^t)}{\hat{\rho}}, \tag{7}$$

*where*

$$\hat{\rho} := \min\{\rho^t\}_{t=1}^{T}, \ \rho^t = \left(1 - 8(1 - \eta_l R)^2 - 120R^2 L^2 \eta_l^2 (\eta_l^2 + \tau_*^t W)\right),$$

$$\bar{\beta} := \frac{1}{T}\sum_{t=1}^{T} \beta^t, \ \beta^t = 10\left(R\eta_l^2 L\sigma_l^2 + 6R^2\eta_l^2 L\sigma_g^2\right)\tau_*^t W, \ \tau_*^t := (\alpha_*^t(N-K) + K)/K,$$

$$\epsilon = 8\left((1 - \eta_l R)^2 + 15R^2\eta_l^4 L^2\right)\sigma_g^2 + 2R\eta_l^2\left(10\eta_l^2 L^2 + 1\right)\sigma_l^2.$$

*Notably, the brief notation $\tau_*^t$ denotes the benefits of utilizing optimal sampling respecting the communication budget $K$. And, $\rho^t, \beta^t$, and $\epsilon$ involve the impacts of local drifts and data heterogeneity in FL.*

**Convergence rate.** The first term in equation 7 indicates the convergence rate of always utilizing the optimal client sampling in Algorithm 1. The results are subjected to data heterogeneity, local drifts, and the optimal factor $\alpha_*$. In practice, the first term can be further minimized by designing a local learning rate $\eta_l$, applying local regularization objectives (Acar et al., 2021; Reddi et al., 2020; Karimireddy et al., 2020), or conducting momentum techniques (Zeng et al., 2023c; Acar et al., 2021). To preserve the generality, we report the original form of convergence rate without minimizing the rate via setting the local hyperparameters. Ideally, the convergence rate $\mathcal{O}(1/\sqrt{T})$ matches the best speed from recent works (Chen et al., 2020; Gu et al., 2021; Jhunjhunwala et al., 2022).

**Sampling impacts.** We use the optimal factor $\alpha_*$ (Definition 4.2) to measure the best convergence rate of using optimal sampling in FedAvg. Then, we decouple the sampling quality function $Q(S^t)$ from primary convergence rates. Concretely, the bound of the second term in equation 7 is related to the performance of the client sampler in FL. Therefore, minimizing the cumulative sampling quality over federated optimization iteration directly enhances optimization quality, which this paper focuses on.

## 5 Theories of the K-Vib Sampler

In this section, we introduce the theoretical design of the K-Vib sampler for federated client sampling. The adaptive sampling objective aligns with the online variance reduction (Salehi et al., 2017; Borsos et al., 2018; El Hanchi & Stephens, 2020) tasks in stochastic optimization. The difference is that we solve the problem in the scenario of FL using ISP, which induces the constraints on sampling probability given in Remark 2.1.

---

**Algorithm 2** K-Vib Sampler

---
**Require:** $N$, $K$, $T$, $\gamma$, and $\theta$.
 1: Initialize client feedback storage $\omega(i) = 0$ for all $i \in [N]$.
 2: **for** time $t$ in $[T]$ **do**
 3:     $\boldsymbol{p}_i^t \propto \sqrt{\omega(i) + \gamma}$                    ▷ by Lemma 5.1
 4:     $\tilde{\boldsymbol{p}}_i^t \leftarrow (1 - \theta) \cdot \boldsymbol{p}_i^t + \theta \frac{K}{N}$, for all $i \in [N]$
 5:     Draw $S^t \sim \tilde{\boldsymbol{p}}^t$                    ▷ ISP
 6:     Receive feedbacks $\pi_t(i)$, and update $\omega(i) \leftarrow \omega(i) + \pi_t^2(i)/\tilde{\boldsymbol{p}}_i^t$ for $i \in S^t$
 7: **end for**

---

### 5.1 Adaptive Client Sampling as Online Optimization

For enhancing federated optimization, our goal is to minimize the cumulative sampling quality $\sum_{t=1}^{T} Q(S^t)$ for achieving tighter convergence bound equation 7. To this end, we first investigate the equation 6 at round $t$ by decomposing to obtain:

$$
\begin{aligned}
Q(S^t) &= \mathbb{E}\left[ \left\| \left( \sum_{i \in S^t} \frac{\boldsymbol{\lambda}_i \boldsymbol{g}_i^t}{\boldsymbol{p}_i^t} - \sum_{i=1}^{N} \boldsymbol{\lambda}_i \boldsymbol{g}_i^t \right) - \left( \sum_{i \in S_*^t} \frac{\boldsymbol{\lambda}_i \boldsymbol{g}_i^t}{\boldsymbol{p}_i^*} - \sum_{i=1}^{N} \boldsymbol{\lambda}_i \boldsymbol{g}_i^t \right) \right\|^2 \right] \\
&\le \mathbb{E}\left[ \left\| \sum_{i \in S^t} \frac{\boldsymbol{\lambda}_i \boldsymbol{g}_i^t}{\boldsymbol{p}_i^t} - \sum_{i=1}^{N} \boldsymbol{\lambda}_i \boldsymbol{g}_i^t \right\|^2 \right] - \mathbb{E}\left[ \left\| \sum_{i \in S_*^t} \frac{\boldsymbol{\lambda}_i \boldsymbol{g}_i^t}{\boldsymbol{p}_i^*} - \sum_{i=1}^{N} \boldsymbol{\lambda}_i \boldsymbol{g}_i^t \right\|^2 \right] \\
&= \sum_{i=1}^{N} \frac{\boldsymbol{\lambda}_i^2 \|\boldsymbol{g}_i^t\|^2}{\boldsymbol{p}_i^t} - \sum_{i=1}^{N} \frac{\boldsymbol{\lambda}_i^2 \|\boldsymbol{g}_i^t\|^2}{\boldsymbol{p}_i^*},
\end{aligned}
$$

where the second inequality uses the fact that $(a - b)^2 \le a^2 - b^2$ when $b \le a$.

Then, we model the client sampling objective as an online convex optimization problem (Salehi et al., 2017; Borsos et al., 2018; El Hanchi & Stephens, 2020). Concretely, we define the feedback from clients as $\pi_t(i) := \boldsymbol{\lambda}_i \|\boldsymbol{g}_i^t\|$ and the cost function $\ell_t(\boldsymbol{p}) := \sum_{i=1}^{N} \frac{\pi_t(i)^2}{\boldsymbol{p}_i}$ as a online convex optimization task[1] respecting sampling probability $\boldsymbol{p}$. Online convex optimization minimizes the *dynamic* regret:

$$
\frac{1}{T} \sum_{t=1}^{T} Q(S^t) \le \frac{1}{T} \text{Regret}_D(T) := \frac{1}{T} \left( \sum_{t=1}^{T} \ell_t(\boldsymbol{p}^t) - \sum_{t=1}^{T} \min_{\boldsymbol{p}} \ell_t(\boldsymbol{p}) \right). \tag{8}
$$

**What does *regret* measure?** *Regret* measures the cumulative discrepancy of applied sampling probability and the *dynamic* optimal Oracle. In Theorem 4.1, we decomposed the cumulative sampling quality as an error term. And, the upper bound of cumulative sampling quality is given by the *regret*. According to equation 3, the ISP induces a tighter regret. Minimizing the upper bound equation 8 can devise sampling probability for ISP to provide a tighter bound for applied FL.

To this end, we are to build an efficient sampler that outputs an exemplary sequence of independent sampling distributions $\{\boldsymbol{p}^t\}_{t=1}^{T}$ such that $\lim_{T \to \infty} \text{Regret}_D(T)/T = 0$. Our upper bound depends on the constraints of the independent sampling procedure on $\boldsymbol{p}$.

### 5.2 Analyzing the Best Fixed Probability

In the federated optimization process, the local updates $\boldsymbol{g}^t$ change, making it challenging to directly bound the cumulative discrepancy between the sampling probability and the dynamic optimal probability. Consequently,

---

[1]Please distinguish the online cost function $\ell_t(\cdot)$ from local empirical loss of client $f_i(\cdot)$ and global loss function $f(\cdot)$. While $\ell_t(\cdot)$ is always convex, $f(\cdot)$ and $f_i(\cdot)$ can be non-convex.

we explore the advantages of employing the best-fixed probability instead. We decompose the equation 8:

$$\text{Regret}_D(T) = \underbrace{\sum_{t=1}^{T} \ell_t(\boldsymbol{p}^t) - \min_{\boldsymbol{p}} \sum_{t=1}^{T} \ell_t(\boldsymbol{p})}_{\text{Regret}_S(T)} + \underbrace{\min_{\boldsymbol{p}} \sum_{t=1}^{T} \ell_t(\boldsymbol{p}) - \sum_{t=1}^{T} \min_{\boldsymbol{p}} \ell_t(\boldsymbol{p})}_{T_{\text{BFP}}}. \tag{9}$$

**Remark.** The static regret $\text{Regret}_S(T)$ denotes the cumulative online loss gap between an applied sequence of probabilities and the *best-fixed* probability in hindsight. The second term indicates the cumulative loss gap between the best-fixed probability in hindsight and the optimal probabilities. We are to bound the terms respectively.

Our analyses rely on a mild assumption of the convergence status of the federated optimization process that sampling methods are applied (Wang et al., 2021). Notably, stochastic optimization (Salehi et al., 2017; Duchi et al., 2011; Boyd et al., 2004) and federated optimization algorithms (Reddi et al., 2020; Wang et al., 2020; Li et al., 2019) typically achieve a sub-linear convergence speed $\mathcal{O}(1/\sqrt{T})$ at least. Therefore, we assume feedback function related to local objective $f_i(\cdot), \forall i \in [N]$ satisfies:

**Assumption 5.1** (Convergence of applied federated optimization). *Our assumptions rely on the convergence performance of federated learning algorithms, i.e., the decaying speed of feedback functions. Let $\Pi_t := \sum_{i=1}^{N} \pi_t(i)$ denote the sum of feedback. Rely on the convergence behavior of an optimization process, we denote the convergence results $\pi_*(i) := \lim_{t \to \infty} \pi_t(i)$, $\Pi_* := \sum_{i=1}^{N} \pi_*(i)$, $\forall i \in [N]$. And, we have $\sum_{t=1}^{T} \Pi_t \geq \Pi_*, V_T(i) = \sum_{t=1}^{T} \left( \pi_t(i) - \pi_*(i) \right)^2, \forall T \geq 1,$. Besides, we denote the largest feedback with $G$, i.e., $\pi_t(i) \leq G, \forall t \in [T], i \in [N]$.*

*As we discussed above the sub-linear convergence speed $\mathcal{O}(1/\sqrt{T})$ can be obtained by general nonconvex federated learning algorithms. We assume that $|\pi_t(i) - \pi_*(i)| \leq \mathcal{O}(1/\sqrt{t})$, and hence implies $V_T(i) \leq \mathcal{O}(\log(T))$. In general, it also covers better optimization problems implying a tighter upper bound for $|\pi_t(i) - \pi_*(i)|$ (e.g., strongly convex and convex federated learning problems). In this paper, the above assumptions guarantee the regret concerning a basic convergence speed of applied FL algorithms, with an additional cost of $\tilde{\mathcal{O}}(\sqrt{T})$.*

**Remark 5.1** (Restricts of Assumption 5.1). *The assumption mentioned above expects that the local feedback sequence of each client will monotonically decrease in the applied federated learning procedures. However, extreme data heterogeneity across clients may result in non-decaying local feedback from some clients, leading to variance in subsequent analysis. We discuss feasible solutions to ensure this assumption in the Limitation 7.*

Importantly, the $G$ denotes the largest feedback during the applied optimization process, instead of assuming bounded gradients. Then, we bound the second term of equation 9 below:

**Theorem 5.1** (Bound of best fixed probability). *Under Assumptions 5.1, sampling a batch of clients with an expected size of $K$, and for any $i \in [N]$ denote $V_T(i) = \sum_{t=1}^{T} \left( \pi_t(i) - \pi_*(i) \right)^2 \leq \mathcal{O}(\log(T))$. For any $T \geq 1$, the averaged hindsight gap admits,*

$$T_{BFP} \leq \frac{T}{K} \left( \sum_{i=1}^{N} \sqrt{\frac{V_T(i)}{T}} \right) \left( 2\Pi_* + \sum_{i=1}^{N} \sqrt{\frac{V_T(i)}{T}} \right).$$

*Proof of sketch.* This bound can be directly proved to solve the convex optimization problem respectively. Please see Appendix D.1 for details. $\square$

Theorem 5.1 indicates a fast convergence of federated optimization induces a lower bound of $V_T(i)$, yielding a tighter regret. As the hindsight bound vanishes with an appropriate FL solver, our objective turns to devise a $\{\boldsymbol{p}_1, \ldots, \boldsymbol{p}_T\}$ that bounds the static regret $\text{Regret}_S(T)$ in equation 9.

### 5.3 Upper Bound of Static Regret

We utilize the classic follow-the-regularized-leader (FTRL) (Shalev-Shwartz et al., 2012; Kalai & Vempala, 2005; Hazan, 2012) framework to design a stable sampling probability sequence, which is formed at time $t$:

$$\boldsymbol{p}^t := \arg\min_{\boldsymbol{p}} \left\{ \sum_{i=1}^{N} \frac{\pi_{1:t-1}^2(i) + \gamma}{\boldsymbol{p}_i} \right\}, \tag{10}$$

where the regularizer $\gamma$ ensures that the distribution does not change too much and prevents assigning a vanishing probability to any clients. It also ensures a minimum sampling probability $p_{\min}$ for some clients. Therefore, we have the closed-form solution as shown below:

**Lemma 5.1** (Solution to equation 10). *Letting $\boldsymbol{a}_i^t = \pi_{1:t-1}^2(i) + \gamma$ and $0 < \boldsymbol{a}_1^t \le \boldsymbol{a}_2^t \le \cdots \le \boldsymbol{a}_N^t$ and $0 < K \le N$, we have*

$$\boldsymbol{p}_i^t = \begin{cases} 1, & \text{if } i \ge l_2, \\ z_t \frac{\sqrt{\boldsymbol{a}_i^t}}{c_t}, & \text{if } i \in (l_1, l_2), \\ p_{min}, & \text{if } i \le l_1, \end{cases} \tag{11}$$

*where $c_t = \sum_{i \in (l_1, l_2)} \sqrt{\boldsymbol{a}_i^t}$, $z_t = K - (N - l_2) + l_1 \cdot p_{min}$ and the $1 \le l_1 \le l_2 \le N$, which satisfies that $\forall i \in (l_1, l_2)$,*

$$\frac{p_{min} \cdot \sum_{l_1 < i < l_2} \boldsymbol{a}_i^t}{z_t} < \boldsymbol{a}_i^t < \frac{\sum_{l_1 < i < l_2} \boldsymbol{a}_i^t}{z_t}.$$

**Remark.** Compared with vanilla optimal sampling probability in equation 4, our sampling probability especially guarantees a minimum sampling probability $p_{\min}$ on the clients with lower feedback. This probability encourages the exploration of the FL system and prevents the case that some clients are never sampled. Besides, the minimum sampling probability $p_{\min}$ is determined by the $\gamma$ and the cumulative feedback from clients during training. For $t = 1, \dots, T$, if applied sampling probability follows Lemma 5.1 with a proper $\gamma$, we guarantee that $\text{Regret}_S(T)/T \le \mathcal{O}(1/\sqrt{T})$, as proved in Appendix D.2.

However, under practical constraints, the server only has access to the feedback information from the past sampled clients. Hence, equation 11 can not be computed accurately. Inspired by Borsos et al., 2018, we construct an additional estimate of the true feedback for all clients denoted by $\tilde{\boldsymbol{p}}$ and let $S^t \sim \tilde{\boldsymbol{p}}^t$. Concretely, $\tilde{\boldsymbol{p}}$ is mixed by the original estimator $\boldsymbol{p}^t$ with a static distribution. Let $\theta \in [0, 1]$, we have

$$\text{Mixing strategy:} \qquad \tilde{\boldsymbol{p}}^t = (1 - \theta)\boldsymbol{p}^t + \theta \frac{K}{N}, \tag{12}$$

where $\tilde{\boldsymbol{p}}^t \ge \theta \frac{K}{N}$, and hence $\tilde{\pi}_t^2(i) \le \pi_t^2(i) \cdot \frac{N}{\theta K} \le G^2 \cdot \frac{N}{\theta K}$.

Analogous to regularizer $\gamma$, the mixing strategy guarantees the least probability that any clients be sampled, thereby encouraging exploration. We present the expected regret bound of the sampling with mixed probability and the K-Vib sampler outlined in Algorithm 2 with theoretical guarantee in Theorem 5.2.

**Theorem 5.2** (Static expected regret with partial feedback). *Under Assumptions 5.1, sampling $S^t \sim \tilde{\boldsymbol{p}}^t$ with $\mathbb{E}[|S^t|] = K$ for all $t = 1, \dots, T$, and letting $\gamma = G^2 \frac{N}{K\theta}, \theta = (\frac{N}{TK})^{1/3}$ with $T \cdot K \ge N$, we obtain the expected regret*

$$\mathbb{E}\left[\text{Regret}_S(T)\right] \le \tilde{\mathcal{O}}\left(N^{\frac{1}{3}} T^{\frac{2}{3}} / K^{\frac{4}{3}}\right), \tag{13}$$

*where $\tilde{\mathcal{O}}$ hides the logarithmic factors.*

*Proof of sketch.* Denoting $\{\pi_t(i)\}_{i \in S^t}$ as partial feedback from sampled points, it incurs

$$\tilde{\pi}_t^2(i) := \frac{\pi_t^2(i)}{\tilde{\boldsymbol{p}}_i^t} \cdot \mathbb{I}_{i \in S^t}, \text{and } \mathbb{E}[\tilde{\pi}_t^2(i)|\tilde{\boldsymbol{p}}_i^t] = \pi_t^2(i), \forall i \in [N].$$

Analogous to equation 8, we define modified cost functions and their unbiased estimates:

$$\tilde{\ell}_t(\boldsymbol{p}) := \sum_{i=1}^{N} \frac{\tilde{\pi}_t^2(i)}{\boldsymbol{p}_i}, \text{and } \mathbb{E}[\tilde{\ell}_t(\boldsymbol{p})|\tilde{\boldsymbol{p}}^t, \ell_t] = \ell_t(\boldsymbol{p}).$$

Our hyperparameters $\gamma, \theta$ are independent. The $\gamma$ is set to guarantee the stability of probability sequence in equation 50. The $\theta$ is set to optimize the final upper bound. Relying on the additional estimates, we have the full cumulative feedback in expectation. In detail, we provide regret bound $\text{Regret}_S(T)$ by directly using Lemma 5.1 in Appendix D.2. Analogously, we can extend the mixed sampling probability $\tilde{\boldsymbol{p}}^t$ to derive the expected regret bound $\mathbb{E}[\text{Regret}_S(T)]$ given in Appendix D.3. □

**Summary.** The K-Vib sampler can work with a federated optimization process providing unbiased full result estimates. Comparing with previous regret bound $\tilde{\mathcal{O}}\left(N^{\frac{1}{3}}T^{\frac{2}{3}}\right)$ (Borsos et al., 2018) and $\mathcal{O}\left(N^{\frac{1}{3}}T^{\frac{2}{3}}\right)$ (El Hanchi & Stephens, 2020), it implements a linear speed up with communication budget $K$. This advantage relies on a tighter formulation of variance obtained via the ISP. For computational complexity, the primary cost involves sorting the cumulative feedback sequence $\{\omega(i)\}_{i=1}^N$ in Algorithm 2. This sorting operation can be performed efficiently with an adaptive sorting algorithm (Estivill-Castro & Wood, 1992), resulting in a time complexity of at most $\mathcal{O}(N \log N)$.

## 6 Experiments

This section evaluates the convergence benefits of utilizing FL client samplers. Our experiment setup aligns with previous works (Li et al., 2020b; Chen et al., 2020).

**Datasets.** The data distribution across clients is shown in Figure 2. We evaluate the theoretical results via experiments on Synthetic datasets, where the data are generated from Gaussian distributions (Li et al., 2020b) and the model is logistic regression $f(\boldsymbol{x}) = \arg\max(W^T\boldsymbol{x} + \boldsymbol{b})$. We generate $N = 100$ clients of each has a synthetic dataset, where the size of each dataset follows the power law. Besides, we evaluate the proposed sampler on the Federated EMNIST (FEMNIST) from LEAF (Caldas et al., 2018). Following Chen et al., 2020, the FEMNIST tasks involve three degrees of unbalanced level (Chen et al., 2020) as shown in Appendix, Figure 2, including FEMNIST v1 (2,231 clients in total, 10% clients hold 82% training images), FEMNIST v2 (1,231 clients in total, 20% client hold 90% training images) and FEMNIST v3 (462 clients in total, 50% client hold 98% training images). We use the same CNN model in (McMahan et al., 2017).

**Baselines.** We demonstrate our improvement by comparison with the uniform sampling and other adaptive unbiased samplers including Multi-armed Bandit Sampler (Mabs) (Salehi et al., 2017), Variance Reducer Bandit (Vrb) (Borsos et al., 2018) and Avare (El Hanchi & Stephens, 2020). We run experiments with the same random seed and vary the seeds across five independent runs. We present the mean performance (solid lines) with the standard deviation (error bars).

**Hyperparameters.** We run $T = 500$ round for all tasks and use vanilla SGD optimizers with constant step size for both clients and the server, with $\eta_g = 1$. To ensure a fair comparison, we set the hyperparameters of all samplers to the optimal values prescribed in their respective original papers, and the choice of hyperparameters is detailed in the Appendix. For the Synthetic dataset task, We set local learning rate $\eta_l = 0.02$, local epoch 1, and batch size 64. For FEMNIST tasks, we set batch size 20, local epochs 3, $\eta_l = 0.01$, and $K = 111, 62, 23$ as 5% of total clients.

### 6.1 Main Results

We compare convergence performance with baselines on Synthetic tasks and FEMNIST tasks. We observe that K-Vib outperforms baselines with a faster convergence speed, and the degree of convergence benefits depends on the experimental settings. The results are shown in Figure 3 and Figure 4 respectively.

In Figure 3, we show the action of all samplers on three metrics. Concretely, the K-Vib implements a lower curve of regret in comparison with baselines. Hence, it creates a better estimate with lower variance for

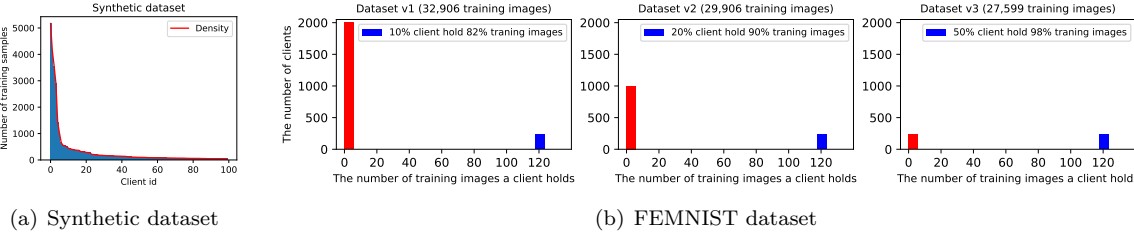

(a) Synthetic dataset                    (b) FEMNIST dataset

Figure 2: Distributed of federated datasets.

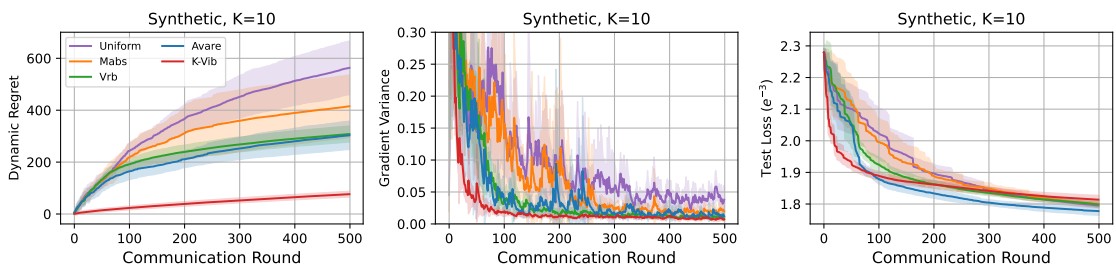

Figure 3: Evaluation of FL samplers on dynamic regret equation 8, gradient variance equation 2, and test loss.

global model updating. Connecting with Theorem 4.1, FedAvg with K-Vib achieves a faster convergence speed as shown in the loss curves.

In Figure 4, the variance of data quantity decreased from FEMNIST v1 to FEMNIST v3. We observe that the FedAvg with the K-Vib sampler converges about $3\times$ faster than baseline when achieving $75\%$ accuracy in FEMINIST v1 and $2\times$ faster in FEMINIST v2. At early rounds, the global estimates provided by naive independent sampling are better as demonstrated in Lemma 2.1, it induces faster convergence by Theorem 4.1. Meanwhile, the K-Vib sampler further enlarges the convergence benefits by solving an online variance reduction task. Hence, it maintains a fast convergence speed. For baseline methods, we observe that the Vrb and Mabs do not outperform the uniform sampling in the FEMNIST task due to the large number of clients and large data quantity variance. In contrast, the Avare sampler fastens the convergence curve after about 150 rounds of exploration in the FEMNIST v1 and v2 tasks. On the FEMNIST v3 task, the Avare sampler shows no clear improvement in the convergence curve, while the K-Vib sampler still implements marginal improvements. Horizontally comparing the results, we observe that the curve discrepancy between K-Vib and baselines is the largest in FEMNIST v1. And, the discrepancy narrows with the decrease of data variance across clients. It indicates that the K-Vib sampler works better in the cross-device FL system with a large number of clients and data variance.

## 6.2   Ablation Study

**Speed up with increasing $K$.** The main advantage of the K-Vib sampler is that the sampling quality is proportional to the communication budget $K$. We present Figure 5 to prove the linear speed up in Theorem 5.2. In detail, with the increase of budget $K$, the performance of the K-Vib sampler with regret metric is reduced significantly. Due to page limitation, we provide further illustration examples of other baselines in the same metric in Appendix Figure 7, where we show that the regret bound of baselines methods are not reduced with increasing communication budget $K$. The results demonstrate our unique improvements in theories.

**Sensitivity to regularizer $\gamma$.** Figure 6 reveals the effects of regularization $\gamma$ in Algorithm 2. The regret slightly changes with different $\gamma$. The variance reduction curves remain stable, indicating the K-Vib

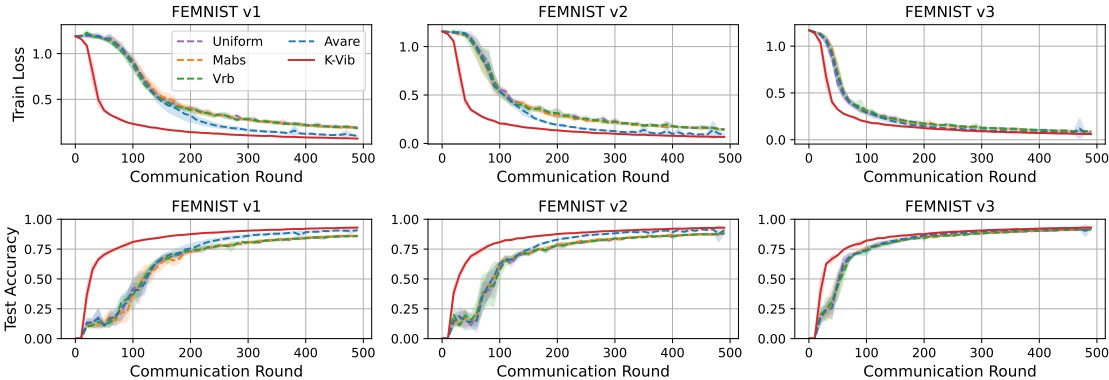

Figure 4: Training loss and test accuracy of FedAvg with different unbiased samplers.

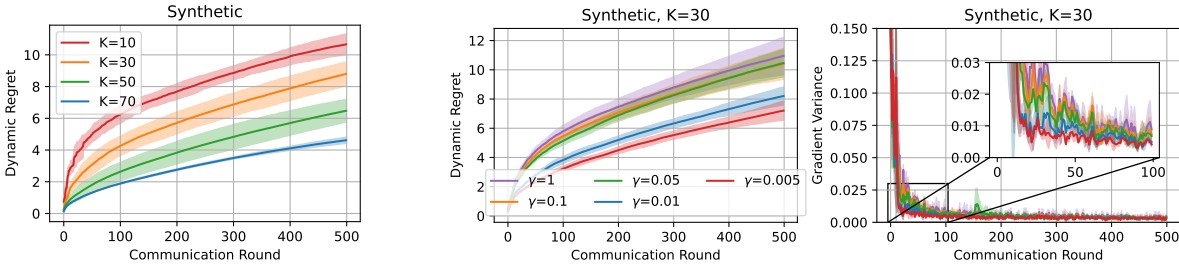

Figure 5: Regret improvement with $K$.

Figure 6: Sensitivity to regularization $\gamma$.

sampler is not sensitive to $\gamma$. This is because the regularizer $\gamma$ only decides the minimum probability in solution equation 11.

## 7 Discussion & Conclusion

**Discussion on additional hyperparameters $\gamma$ and $\theta$.** In this paper, we set $\theta = (\frac{N}{TK})^{\frac{1}{3}}$ and $\gamma \approx G^2 \frac{N}{\theta K}$, which aligns with our theoretical analysis. Concretely, we guarantee the stability of designed probability via $\gamma$ in equation 55. In practice, we use the mean value of first-round client feedback as a naive estimate of $G$. This is because the first-round feedback is typically the largest during FL training based on Assumption 5.1. Besides, we optimize the final regret bound in equation 56 respecting $\theta$ and set $\theta = (\frac{N}{TK})^{\frac{1}{3}}$. This hyperparameter tuning experience can be applied in future applications.

**Extension & Limitation.** Our theoretical findings can be extended to general applications that involve estimating global results with partial information. Additionally, our extension of independent sampling can be applied to previous works employing random sampling. Besides, the global estimate variance in FL also comes from the data heterogeneity issues, which may incur non-decaying local feedback, hence breaking the Assumption 5.1. This can be addressed with client clustering techniques (Ghosh et al., 2020; Ma et al., 2022; Zeng et al., 2023a), analogous to previously cluster sampling works (Fraboni et al., 2021; Song et al., 2023). Besides, we can replace FedAvg with more stable FedAvg variants (Sun et al., 2024; Zeng et al., 2023c).

In conclusion, our study provides a thorough examination of FL frameworks utilizing unbiased client sampling techniques from an optimization standpoint. Our findings highlight the importance of designing unbiased sampling probabilities for the independent sampling procedure to enhance the efficiency of FL. Building upon this insight, we further extend the range of adaptive sampling techniques and achieve substantial improvements. We are confident that our work will contribute to the advancement of client sampling techniques in FL, making them more applicable and beneficial in various practical scenarios.

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

# Appendices

# A    Related Work

Our paper contributes to the literature on the importance sampling in stochastic optimization, online convex optimization, and client sampling in FL.

**Importance Sampling.** Importance sampling is a non-uniform sampling technique widely used in stochastic optimization (Katharopoulos & Fleuret, 2018) and coordinate descent (Richtárik & Takáč, 2016a). Zhao & Zhang (2015); Needell et al. (2014) connects the variance of the gradient estimates and the optimal sampling distribution is proportional to the per-sample gradient norm. The insights of sampling and optimization quality can be transferred into federated client sampling, as we summarised in the following two topics.

**Online Variance Reduction.** Our paper addresses the topic of online convex optimization for reducing variance. Variance reduction techniques are frequently used in conjunction with stochastic optimization algorithms (Defazio et al., 2014; Johnson & Zhang, 2013) to enhance optimization performance. These same variance reduction techniques have also been proposed to quicken federated optimization (Dinh et al., 2020; Malinovsky et al., 2022). On the other hand, online learning (Shalev-Shwartz et al., 2012) typically employs an exploration-exploitation paradigm to develop decision-making strategies that maximize profits. Although some studies have considered client sampling as a multi-armed bandit problem, they have only provided limited theoretical results (Kim et al., 2020; Cho et al., 2020a; Yang et al., 2021). In an intriguing combination, certain studies (Salehi et al., 2017; Borsos et al., 2018; 2019) have formulated data sampling in stochastic optimization as an online learning problem. These methods were also applied to client sampling in FL by treating each client as a data sample in their original problem (Zhao et al., 2021a; El Hanchi & Stephens, 2020).

**Client Sampling in FL.** Client sampling methods in FL fall under two categories: biased and unbiased methods. Unbiased sampling methods ensure objective consistency in FL by yielding the same expected value of results as global aggregation with the participation of all clients. In contrast, biased sampling methods converge to arbitrary sub-optimal outcomes based on the specific sampling strategies utilized. Additional discussion about biased and unbiased sampling methods is provided in Appendix E.2. Recent research has focused on exploring various client sampling strategies for both biased and unbiased methods. For instance, biased sampling methods involve sampling clients with probabilities proportional to their local dataset size (McMahan et al., 2017), selecting clients with a large update norm with higher probability (Chen et al., 2020), choosing clients with higher losses (Cho et al., 2020b), and building a submodular maximization to approximate the full gradients (Balakrishnan et al., 2022). Meanwhile, several studies (Chen et al., 2020; Cho et al., 2020b) have proposed theoretically optimal sampling methods for FL utilizing the unbiased sampling framework, which requires all clients to upload local information before conducting sampling action. Moreover, cluster-based sampling (Fraboni et al., 2021; Xu et al., 2021; Shen et al., 2022) relies on additional clustering operations where the knowledge of utilizing client clustering can be transferred into other client sampling techniques.

# B    Useful Lemmas and Corollaries

## B.1    Auxiliary Lemmas

**Lemma B.1** (Lemma 13, Borsos et al., 2018). *For any sequence of numbers $c_1, \ldots, c_T \in [0, 1]$ the following holds:*

$$\sum_{t=1}^{T} \frac{c_t^4}{(c_{1:t}^2)^{3/2}} \leq 44,$$

*where $c_{1:t} = \sum_{\tau=1}^{t} c_\tau$.*

**Lemma B.2.** *For an arbitrary set of $n$ vectors $\{\boldsymbol{a}_i\}_{i=1}^{n}, \boldsymbol{a}_i \in \mathbb{R}^d$,*

$$\left\| \sum_{i=1}^{n} \mathbf{a}_i \right\|^2 \leq n \sum_{i=1}^{n} \|\mathbf{a}_i\|^2. \tag{14}$$

**Lemma B.3.** *For random variables $z_1, \ldots, z_n$, we have*

$$\mathbb{E}\left[\|z_1 + \ldots + z_n\|^2\right] \leq n\mathbb{E}\left[\|z_1\|^2 + \ldots + \|z_n\|^2\right]. \tag{15}$$

**Lemma B.4.** *For independent, mean 0 random variables $z_1, \ldots, z_n$, we have*

$$\mathbb{E}\left[\|z_1 + \ldots + z_n\|^2\right] = \mathbb{E}\left[\|z_1\|^2 + \ldots + \|z_n\|^2\right]. \tag{16}$$

## B.2 Arbitrary Sampling

In this section, we summarize the arbitrary sampling techniques and present key lemmas used in this paper. The arbitrary sampling is mainly used either for generating mini-batches of samples in stochastic algorithms (Chambolle et al., 2018; Richtárik & Takáč, 2016a) or for coordinate descent optimization (Qu & Richtárik, 2016). In contrast, we explain the background in the context of federated optimization.

In detail, let $S$ denote a sampling, which is a random set-valued mapping with values in $2^{[N]}$, where $[N] := \{1, 2, \ldots, N\}$. An arbitrary sampling $S$ is generated by assigning probabilities to all $2^N$ subsets of $[N]$, which associates a *probability matrix* $\mathbf{P} \in \mathbb{R}^{N \times N}$ defined by

$$\mathbf{P}_{ij} := \text{Prob}(\{i, j\} \subseteq S).$$

Thus, the *probability vector* $p = (\boldsymbol{p}_1, \ldots, p_N) \in \mathbb{R}^N$ is composed of the diagonal entries of $\mathbf{P}$, and $\boldsymbol{p}_i := \text{Prob}(i \in S)$. Furthermore, we say that $S$ is *proper* if $\boldsymbol{p}_i > 0$ for all $i$. Thus, it incurs that

$$K := \mathbb{E}[|S|] = \text{Trace}(\mathbf{P}) = \sum_{i=1}^{N} \boldsymbol{p}_i.$$

The definition of sampling can be naively transferred to the context of federated client sampling. We refer to $K$ as the expected number of sampled clients per round in FL. The following lemma plays a key role in our problem formulation and analysis.

**Lemma B.5** (Generalization of Lemma 1 Horváth & Richtárik (2019)). *Let $\boldsymbol{a}_1, \boldsymbol{a}_2, \ldots, \boldsymbol{a}_N$ be vectors in $\mathbb{R}^d$ and let $\bar{\boldsymbol{a}} = \sum_{i=1}^{N} \boldsymbol{\lambda}_i \boldsymbol{a}_i$ be their weighted average. Let $S$ be a proper sampling. Assume that there is $\boldsymbol{v} \in \mathbb{R}^N$ such that*

$$\mathbf{P} - pp^t \preceq \boldsymbol{Diag}(\boldsymbol{p}_1\boldsymbol{v}_1, \boldsymbol{p}_2\boldsymbol{v}_2, \ldots, \boldsymbol{p}_N\boldsymbol{v}_N). \tag{17}$$

*Then, we have*

$$\mathbb{E}_{S \sim p}\left[\left\|\sum_{i \in S} \frac{\boldsymbol{\lambda}_i \boldsymbol{a}_i}{\boldsymbol{p}_i} - \bar{\boldsymbol{a}}\right\|^2\right] \leq \sum_{i=1}^{N} \boldsymbol{\lambda}_i^2 \frac{\boldsymbol{v}_i}{\boldsymbol{p}_i} \|\boldsymbol{a}_i\|^2, \tag{18}$$

*where the expectation is taken over sampling $S$. Whenever equation 17 holds, it must be the case that*

$$\boldsymbol{v}_i \geq 1 - \boldsymbol{p}_i.$$

*Moreover, The random sampling admits $\boldsymbol{v}_i = \frac{N-K}{N-1}$. The independent sampling admits $\boldsymbol{v}_i = 1 - \boldsymbol{p}_i$ and makes equation 18 hold as equality.*

*Proof.* Let $\mathbb{I}_{i \in S} = 1$ if $i \in S$ and $\mathbb{I}_{i \in S} = 0$ otherwise. Similarly, let $\mathbb{I}_{i,j \in S} = 1$ if $i \in S$ and $\mathbb{I}_{i,j \in S} = 0$ otherwise. Note that $\mathbb{E}[\mathbb{I}_{i \in S}] = \boldsymbol{p}_i$ and $\mathbb{E}[\mathbb{I}_{i,j \in S}] = \mathbf{P}_{ij}$. Then, we compute the mean of estimates $\tilde{\boldsymbol{a}} := \sum_{i \in S} \frac{\boldsymbol{\lambda}_i \boldsymbol{a}_i}{\boldsymbol{p}_i}$:

$$\mathbb{E}[\tilde{\boldsymbol{a}}] = \mathbb{E}[\sum_{i \in S} \frac{\boldsymbol{\lambda}_i \boldsymbol{a}_i}{\boldsymbol{p}_i}] = \mathbb{E}[\sum_{i=1}^{N} \frac{\boldsymbol{\lambda}_i \boldsymbol{a}_i}{\boldsymbol{p}_i} \mathbb{I}_{i \in S}] = \sum_{i=1}^{N} \frac{\boldsymbol{\lambda}_i \boldsymbol{a}_i}{\boldsymbol{p}_i} \mathbb{E}[\mathbb{I}_{i \in S}] = \sum_{i=1}^{N} \boldsymbol{\lambda}_i \boldsymbol{a}_i = \bar{\boldsymbol{a}}.$$

Let $\mathbf{A} = [\boldsymbol{\zeta}_1, \ldots, \boldsymbol{\zeta}_N] \in \mathbb{R}^{d \times N}$, where $\boldsymbol{\zeta}_i = \frac{\boldsymbol{\lambda}_i \boldsymbol{a}_i}{\boldsymbol{p}_i}$, and let $\boldsymbol{e}$ be the vector of all ones in $\mathbb{R}^N$. We now write the variance of $\tilde{\boldsymbol{a}}$ in a form that will be convenient to establish a bound:

$$
\begin{aligned}
\mathbb{E}[\|\tilde{\boldsymbol{a}} - \mathbb{E}[\tilde{\boldsymbol{a}}]\|^2] &= \mathbb{E}[\|\tilde{\boldsymbol{a}}\|^2] - \|\mathbb{E}[\tilde{\boldsymbol{a}}]\|^2 \\
&= \mathbb{E}[\| \sum_{i \in S} \frac{\boldsymbol{\lambda} \boldsymbol{a}_i}{\boldsymbol{p}_i}\|^2] - \|\bar{\boldsymbol{a}}\|^2 \\
&= \mathbb{E}\left[ \sum_{i,j} \frac{\boldsymbol{\lambda}_i \boldsymbol{a}_i^\top}{\boldsymbol{p}_i} \frac{\boldsymbol{\lambda}_j \boldsymbol{a}_j}{\boldsymbol{p}_j} \mathbb{I}_{i,j \in S} \right] - \|\bar{\boldsymbol{a}}\|^2 \\
&= \sum_{i,j} \boldsymbol{p}_{ij} \frac{\boldsymbol{\lambda}_i \boldsymbol{a}_i^\top}{\boldsymbol{p}_i} \frac{\boldsymbol{\lambda}_j \boldsymbol{a}_j}{\boldsymbol{p}_j} - \sum_{i,j} \boldsymbol{\lambda}_i \boldsymbol{\lambda}_j \boldsymbol{a}_i^\top \boldsymbol{a}_j \\
&= \sum_{i,j} \left( \boldsymbol{p}_{ij} - \boldsymbol{p}_i \boldsymbol{p}_j \right) \boldsymbol{\zeta}_i^\top \boldsymbol{\zeta}_j \\
&= \boldsymbol{e}^\top \left( (\mathbf{P} - \boldsymbol{p}\boldsymbol{p}^\top) \circ \mathbf{A}^\top \mathbf{A} \right) \boldsymbol{e}.
\end{aligned}
\tag{19}
$$

Since by assumption we have $\mathbf{P} - \boldsymbol{p}\boldsymbol{p}^\top \preceq \mathbf{Diag}(\boldsymbol{p} \circ \boldsymbol{v})$, we can further bound

$$
\boldsymbol{e}^\top \left( (\mathbf{P} - \boldsymbol{p}\boldsymbol{p}^\top) \circ \mathbf{A}^\top \mathbf{A} \right) \boldsymbol{e} \leq \boldsymbol{e}^\top \left( \mathbf{Diag}(\boldsymbol{p} \circ \boldsymbol{v}) \circ \mathbf{A}^\top \mathbf{A} \right) \boldsymbol{e} = \sum_{i=1}^n \boldsymbol{p}_i \boldsymbol{v}_i \|\boldsymbol{\zeta}_i\|^2.
\tag{20}
$$

To obtain equation 18, it remains to combine equation 20 with equation 19. Since $\mathbf{P} - \boldsymbol{p}\boldsymbol{p}^\top$ is positive semi-definite (Richtárik & Takáč, 2016b), we can bound $\mathbf{P} - \boldsymbol{p}\boldsymbol{p}^\top \preceq N\mathbf{Diag}(\mathbf{P} - \boldsymbol{p}\boldsymbol{p}^\top) = \mathbf{Diag}(\boldsymbol{p} \circ \boldsymbol{v})$, where $\boldsymbol{v}_i = N(1 - \boldsymbol{p}_i)$.

Overall, arbitrary sampling that associates with a probability matrix $\mathbf{P}$ will determine the value of $\boldsymbol{v}$. As a result, we summarize independent sampling and random sampling as follows,

- Consider now the independent sampling,

$$
\mathbf{P} - \boldsymbol{p}\boldsymbol{p}^\top = \begin{bmatrix}
\boldsymbol{p}_1 (1 - \boldsymbol{p}_1) & 0 & \cdots & 0 \\
0 & \boldsymbol{p}_2 (1 - \boldsymbol{p}_2) & \cdots & 0 \\
\vdots & \vdots & \ddots & \vdots \\
0 & 0 & \cdots & \boldsymbol{p}_n (1 - \boldsymbol{p}_n)
\end{bmatrix} = \mathbf{Diag}\left( \boldsymbol{p}_1 \boldsymbol{v}_1, \ldots, \boldsymbol{p}_n \boldsymbol{v}_n \right),
$$

  where $\boldsymbol{v}_i = 1 - \boldsymbol{p}_i$. Therefore, independent sampling always minimizes equation 18, making it hold as equality.

- Consider the random sampling,

$$
\mathbf{P} - \boldsymbol{p}\boldsymbol{p}^\top = \begin{bmatrix}
\frac{K}{N} - \frac{K^2}{N^2} & \frac{K(K-1)}{N(N-1)} & \cdots & \frac{K(K-1)}{N(N-1)} \\
\frac{K(K-1)}{N(N-1)} & \frac{K}{N} & \cdots & \frac{K(K-1)}{N(N-1)} \\
\vdots & \vdots & \ddots & \vdots \\
\frac{K(K-1)}{N(N-1)} & \frac{K(K-1)}{N(N-1)} & \cdots & \frac{K}{N}
\end{bmatrix}.
$$

  As shown in (Horváth & Richtárik, 2019), the standard random sampling admits $\boldsymbol{v}_i = \frac{N-K}{N-1}$ for equation 18.

$\square$

**_Conclusion._** Given probabilities $\boldsymbol{p}$ that defines all samplings $S$ satisfying $\boldsymbol{p}_i = \mathrm{Prob}(i \in S)$, it turns out that the independent sampling (i.e., $\mathbf{P}_{ij} = \mathrm{Prob}(i, j \in S) = \mathrm{Prob}(i \in S)\mathrm{Prob}(j \in S) = \boldsymbol{p}_i \boldsymbol{p}_j$) minimizes the upper bound in equation 18. Therefore, depending on the sampling distribution and method, we can rewrite the equation 18 as follows:

$$\mathbb{V}(S) = \mathbb{E}_{S \sim \boldsymbol{p}}[\| \sum_{i \in S} \frac{\boldsymbol{\lambda}_i \boldsymbol{a}_i}{\boldsymbol{a}_i} - \bar{\boldsymbol{a}} \|^2] = \underbrace{\sum_{i=1}^{N} (1 - \boldsymbol{p}_i) \frac{\boldsymbol{\lambda}_i^2 \|\boldsymbol{a}_i\|^2}{\boldsymbol{p}_i}}_{\text{Independent Sampling}} \leq \underbrace{\frac{N-K}{N-1} \sum_{i=1}^{N} \frac{\boldsymbol{\lambda}_i^2 \|\boldsymbol{a}_i\|^2}{\boldsymbol{p}_i}}_{\text{Random Sampling}}. \tag{21}$$

### B.3 Proof of Solution to Independent Sampling with Minimal Probability

In this section, we present lemmas and their proofs for our theoretical analyses. Our methodology of independent sampling especially guarantees a minimum probability of clients in comparison with Lemma 2.2. Our proof involves a general constraint, which covers Lemma 2.2. Then, we provide several Corollaries B.1 B.2 for our analysis in the next section.

**Lemma B.6.** *Let $0 < \boldsymbol{a}_1 \leq \boldsymbol{a}_2 \leq \cdots \leq \boldsymbol{a}_N$ and $0 < K \leq N$. We consider the following optimization objective with a restricted probability space $\Delta = \{\boldsymbol{p} \in \mathbb{R}^N | p_{min} \leq \boldsymbol{p}_i \leq 1, \sum_{i=1}^{N} \boldsymbol{p}_i = K, \forall i \in [N]\}$ where $p_{min} \leq K/N$,*

$$minimize_{\boldsymbol{p} \in \Delta} \ \Omega(\boldsymbol{p}) = \sum_{i=1}^{N} \frac{\boldsymbol{a}_i^2}{\boldsymbol{p}_i}$$

$$subject \ to \ \sum_{i=1}^{N} \boldsymbol{p}_i = K, \tag{22}$$

$$p_{min} \leq \boldsymbol{p}_i \leq 1, \ i = 1, 2, \ldots, N.$$

*Proof.* We formulate the Lagrangian:

$$\mathcal{L}(p, y, \alpha_1, \ldots, \alpha_N, \beta_1, \ldots, \beta_N) = \sum_{i=1}^{N} \frac{\boldsymbol{a}_i^2}{\boldsymbol{p}_i} + y \cdot \left( \sum_{i=1}^{N} \boldsymbol{p}_i - K \right) + \sum_{i=1}^{N} \alpha_i (p_{\min} - \boldsymbol{p}_i) + \sum_{i=1}^{N} \beta_i (\boldsymbol{p}_i - 1). \tag{23}$$

The constraints are linear and KKT conditions hold. Hence, we have,

$$\boldsymbol{p}_i = \sqrt{\frac{\boldsymbol{a}_i^2}{y - \alpha_i + \beta_i}} = \begin{cases} 1, & \text{if } \sqrt{y} \leq \boldsymbol{a}_i. \\ \sqrt{\frac{\boldsymbol{a}_i^2}{y}}, & \text{if } \sqrt{y} \cdot p_{\min} < \boldsymbol{a}_i < \sqrt{y}, \\ p_{\min}, & \text{if } \boldsymbol{a}_i \leq \sqrt{y} \cdot p_{\min}. \end{cases} \tag{24}$$

Then, we analyze the value of $y$. Letting $l_1 = \left| \{i | \boldsymbol{a}_i \leq \sqrt{y} \cdot p_{\min}\} \right|$, $l_2 = l_1 + |\{\sqrt{y} \cdot p_{\min} < \boldsymbol{a}_i < \sqrt{y}\}|$, $N - l_2 = \left| \{i | \sqrt{y} \leq \boldsymbol{a}_i\} \right|$, and using $\sum_{i=1}^{N} \boldsymbol{p}_i = K$ implies,

$$\sum_{i=1}^{N} \boldsymbol{p}_i = \sum_{i \leq l_1} \boldsymbol{p}_i + \sum_{l_1 < i < l_2} \boldsymbol{p}_i + \sum_{i \geq l_2} \boldsymbol{p}_i = l_1 \cdot p_{\min} + \sum_{l_1 < i < l_2} \sqrt{\frac{\boldsymbol{a}_i^2}{y}} + N - l_2 = K.$$

Arrange the formula, we get

$$\sqrt{y} = \frac{\sum_{l_1 < i < l_2} \boldsymbol{a}_i}{K - N + l_2 - l_1 \cdot p_{\min}}. \tag{25}$$

Moreover, we can plug the results into the objective to get the optimal result:

$$\sum_{i=1}^{N} \frac{\boldsymbol{a}_i^2}{\boldsymbol{p}_i} = \sum_{i \leq l_1} \frac{\boldsymbol{a}_i^2}{\boldsymbol{p}_i} + \sum_{l_1 < i < l_2} \frac{\boldsymbol{a}_i^2}{\boldsymbol{p}_i} + \sum_{i \geq N - l_2} \frac{\boldsymbol{a}_i^2}{\boldsymbol{p}_i}$$

$$= \frac{\sum_{i \leq l_1} \boldsymbol{a}_i^2}{p_{\min}} + \sqrt{y} \left( \sum_{l_1 < i < l_2} \boldsymbol{a}_i \right) + \sum_{i \geq N - l_2} \boldsymbol{a}_i^2 \tag{26}$$

$$= \frac{\sum_{i \leq l_1} \boldsymbol{a}_i^2}{p_{\min}} + \frac{(\sum_{l_1 < i < l_2} \boldsymbol{a}_i)^2}{K - N + (l_2 - l_1 \cdot p_{\min})} + \sum_{i \geq N - l_2} \boldsymbol{a}_i^2,$$

where the $1 \le l_1 \le l_2 \le N$, which satisfies that $\forall i \in (l_1, l_2)$,

$$p_{\min} \cdot \frac{\sum_{l_1 < i < l_2} \boldsymbol{a}_i}{K - N + l_2 - l_1 \cdot p_{\min}} < \boldsymbol{a}_i < \frac{\sum_{l_1 < i < l_2} \boldsymbol{a}_i}{K - N + l_2 - l_1 \cdot p_{\min}}.$$

In short, we note that if let $p_{\min} = 0, l_1 = 0$, the Lemma 2.2 is proved as a special case of equation 26. Besides, we provide further Corollary B.1 and B.2 as preliminaries for further analysis.

**Corollary B.1.** *With $K \cdot a_N \le \sum_{i=1}^{N} \boldsymbol{a}_i$ and $p_{min} = 0$, we have $l_1 = 0, l_2 = N$ for equation 26 and induce*

$$\arg \min \Omega(\boldsymbol{p}^*) = \frac{(\sum_{i=1}^{N} \boldsymbol{a}_i)^2}{K}.$$

**Corollary B.2.** *With $K \cdot a_N \le \sum_{i=1}^{N} \boldsymbol{a}_i$ and $p_{min} > 0$, we have $l_2 = N$ and $l_1$ is the largest integer that satisfies $0 < (K - l_1 \cdot p_{min}) \frac{a_{l_1}}{\sum_{i=l_1}^{N} \boldsymbol{a}_i} < p_{min}$. The optimal value of equation 26 becomes*

$$
\begin{aligned}
\sum_{i=1}^{N} \frac{\boldsymbol{a}_i^2}{\boldsymbol{p}_i} &= \frac{\sum_{i \le l_1} \boldsymbol{a}_i^2}{p_{min}} + \sqrt{y}(\sum_{l_1 < i \le N} \boldsymbol{a}_i) && \triangleright \text{Eq. equation 26, def. in line 2} \\
&= \frac{\sum_{i \le l_1} \boldsymbol{a}_i^2}{p_{min}} + y(K - l_1 p_{min}) && \triangleright \text{Eq. 25, replacing } \sum_{l_1 < i \le N} \boldsymbol{a}_i \\
&\le l_1 y p_{min} + y(K - l_1 p_{min}) && \triangleright \text{Eq. 24, } \boldsymbol{a}_i \le \sqrt{y} \cdot p_{min} \\
&= \frac{(\sum_{i=l_1}^{N} \boldsymbol{a}_i)^2}{(K - l_1 p_{min})^2} \cdot K \le \frac{K(\sum_{i=l_1}^{N} \boldsymbol{a}_i)^2}{(K - N p_{min})^2} \\
&\le \frac{K(\sum_{i=1}^{N} \boldsymbol{a}_i)^2}{(K - N p_{min})^2}.
\end{aligned}
$$

$\square$

## C  Convergence Analyses

### C.1  Sampling and Bounded Local Drift

We start our convergence analysis with a clarification of the concepts of optimal independent sampling. Considering an Oracle always outputs the optimal probabilities $\boldsymbol{p}^*$, we define

$$\delta_*^t := \mathbb{E}\left[\left\|\sum_{i \in S^*} \frac{\boldsymbol{\lambda}_i \boldsymbol{g}_i^t}{\boldsymbol{p}_i^*} - \sum_{i=1}^{N} \boldsymbol{\lambda}_i \boldsymbol{g}_i^t\right\|^2\right] = \mathbb{E}\left[\sum_{i=1}^{N} \frac{1 - \boldsymbol{p}_i^*}{\boldsymbol{p}_i^*}\|\tilde{\boldsymbol{g}}_i^t\|^2\right],$$

where we have $\|\tilde{\boldsymbol{g}}_i^t\|^2 = \|\boldsymbol{\lambda}_i \boldsymbol{g}_i^t\|^2$. Then, we plug the optimal probability in equation 4 into the above equation to obtain

$$\delta_*^t = \mathbb{E}\left[\sum_{i=1}^{N} \frac{1 - \boldsymbol{p}_i^*}{\boldsymbol{p}_i^*}\|\tilde{\boldsymbol{g}}_i^t\|^2\right] = \mathbb{E}\left[\frac{1}{K - (N - l)}\left(\sum_{i=1}^{l}\|\tilde{\boldsymbol{g}}_i^t\|\right)^2 - \sum_{i=1}^{l}\|\tilde{\boldsymbol{g}}_i^t\|^2\right].$$

Using the fact that $K\|\tilde{\boldsymbol{g}}_N^t\| \le \sum_{i=1}^N \|\tilde{\boldsymbol{g}}_i^t\|$, we have

$$\delta_*^t \le \mathbb{E}\left[\frac{1}{K}\left(\sum_{i=1}^N \|\tilde{\boldsymbol{g}}_i^t\|\right)^2 - \sum_{i=1}^N \|\tilde{\boldsymbol{g}}_i^t\|^2\right]$$

$$= \mathbb{E}\left[\frac{1}{K}\left(\sum_{i=1}^N \|\tilde{\boldsymbol{g}}_i^t\|\right)^2 \left(1 - K\frac{\sum_{i=1}^N \|\tilde{\boldsymbol{g}}_i^t\|^2}{\left(\sum_{i=1}^N \|\tilde{\boldsymbol{g}}_i^t\|\right)^2}\right)\right]$$

$$\le \frac{N-K}{NK}\mathbb{E}\left[\left(\sum_{i=1}^N \|\tilde{\boldsymbol{g}}_i^t\|\right)^2\right].$$

To clarify the improvement of utilizing the sampling procedure, we provide two baseline analyses respecting independent sampling and random sampling. For an uniform independent sampling $S^t \sim \mathbb{U}(\boldsymbol{p}_i = \frac{K}{N})$, we have

$$\delta^t := \mathbb{E}\left[\left\|\sum_{i\in S^t}\frac{\boldsymbol{\lambda}_i}{\boldsymbol{p}_i}\boldsymbol{g}_i^t - \sum_{i=1}^N \boldsymbol{\lambda}_i\boldsymbol{g}_i^t\right\|^2\right] = \mathbb{E}\left[\sum_{i=1}^N \frac{1-\frac{K}{N}}{\frac{K}{N}}\|\tilde{\boldsymbol{g}}_i^t\|^2\right] = \frac{N-K}{K}\mathbb{E}\left[\sum_{i=1}^N \|\tilde{\boldsymbol{g}}_i^t\|^2\right]$$

For a uniform random sampling $S^t \sim \mathbb{U}(\boldsymbol{p}_i = \frac{K}{N})$, we have by Lemma 2.2

$$\delta_{\mathbb{U}} := \mathbb{E}\left[\left\|\sum_{i\in S^t}\frac{\boldsymbol{\lambda}_i}{\boldsymbol{p}_i}\boldsymbol{g}_i^t - \sum_{i=1}^N \boldsymbol{\lambda}_i\boldsymbol{g}_i^t\right\|^2\right] \le \frac{N-K}{N-1}\frac{N}{K}\mathbb{E}\left[\sum_{i=1}^N \|\tilde{\boldsymbol{g}}_i^t\|^2\right]. \tag{27}$$

Straightforwardly, we can prove that $\delta^t/\delta_{\mathbb{U}} < 0$, indicating that independent sampling creates better estimates than random sampling.

**Definition C.1** (The optimal factor)**.** *Given an iteration sequence of global model* $\{\boldsymbol{x}^1, \ldots, \boldsymbol{x}^t\}$, *under the constraints of communication budget* $K$ *and local updates statues* $\{\boldsymbol{g}_i^t\}_{i\in[N]}, t \in [T]$, *we define the improvement factor of applying optimal client sampling* $S_*^t \sim \boldsymbol{p}^*$ *comparing uniform sampling* $U^t \sim \mathbb{U}$ *as:*

$$\alpha_*^t = \frac{\mathbb{E}\left[\left\|\sum_{i\in S_*^t}\frac{\boldsymbol{\lambda}_i}{\boldsymbol{p}_i^*}\boldsymbol{g}_i^t - \sum_{i=1}^N \boldsymbol{\lambda}_i\boldsymbol{g}_i^t\right\|^2\right]}{\mathbb{E}\left[\left\|\sum_{i\in U^t}\frac{\boldsymbol{\lambda}_i}{\boldsymbol{p}_i}\boldsymbol{g}_i^t - \sum_{i=1}^N \boldsymbol{\lambda}_i\boldsymbol{g}_i^t\right\|^2\right]},$$

*and optimal* $\boldsymbol{p}^*$ *is computed via equation 4 with* $\{\boldsymbol{g}_i^t\}_{i\in[N]}$.

Putting Equations together induces the improvement factor of optimal independent sampling respecting uniform random sampling:

$$\alpha_*^t := \frac{\delta_*^t}{\delta_{\mathbb{U}}} = \frac{\mathbb{E}\left[\left\|\sum_{i\in S^*}\frac{\boldsymbol{\lambda}_i}{\boldsymbol{p}_i^*}\boldsymbol{g}_i^t - \sum_{i=1}^N \boldsymbol{\lambda}_i\boldsymbol{g}_i^t\right\|^2\right]}{\mathbb{E}\left[\left\|\sum_{i\in S^t}\frac{\boldsymbol{\lambda}_i}{\boldsymbol{p}_i}\boldsymbol{g}_i^t - \sum_{i=1}^N \boldsymbol{\lambda}_i\boldsymbol{g}_i^t\right\|^2\right]}$$

$$\le \frac{(N-1)\mathbb{E}\left[\left(\sum_{i=1}^N \|\tilde{\boldsymbol{g}}_i^t\|\right)^2\right]}{N^2\mathbb{E}\left[\sum_{i=1}^N \|\tilde{\boldsymbol{g}}_i^t\|^2\right]} < \frac{\mathbb{E}\left[\left(\sum_{i=1}^N \|\tilde{\boldsymbol{g}}_i^t\|\right)^2\right]}{N\mathbb{E}\left[\sum_{i=1}^N \|\tilde{\boldsymbol{g}}_i^t\|^2\right]} \le 1. \tag{28}$$

**Lemma C.1** (Upper bound of local drift, Reddi et al., 2020)**.** *Let Assumption 4.2 4.3 hold. For all client* $i \in [N]$ *with arbitrary local iteration steps* $r \in [R]$, *the local drift can be bounded as follows,*

$$\mathbb{E}\left\|\boldsymbol{x}_i^{t,r} - \boldsymbol{x}^t\right\|^2 \le 5R\eta_l^2(\sigma_l^2 + 6R\sigma_g^2 + 6R\left\|\nabla f\left(\boldsymbol{x}^t\right)\right\|^2)$$

*Proof.* For $r \in [R]$, we have

$$\mathbb{E}\left[\left\|\boldsymbol{g}_i^t\right\|^2\right] = \mathbb{E}\left\|\boldsymbol{x}_i^{t,r} - \boldsymbol{x}^t\right\|^2 = \mathbb{E}\left\|\boldsymbol{x}_i^{t,r-1} - \boldsymbol{x}^t - \eta_l \nabla F_i(\boldsymbol{x}_i^{t,r-1})\right\|^2$$

$$= \mathbb{E}\left\|\boldsymbol{x}_i^{t,r-1} - \boldsymbol{x}^t - \eta_l(\nabla F_i(\boldsymbol{x}_i^{t,r-1}) \pm \nabla F_i(\boldsymbol{x}_i^{t,r-1}))\right\|^2$$

$$= \mathbb{E}\left\|\boldsymbol{x}_i^{t,r-1} - \boldsymbol{x}^t - \eta_l \nabla F_i(\boldsymbol{x}_i^{t,r-1}))\right\|^2 + \mathbb{E}\left\|\eta_l\left(\nabla F_i(\boldsymbol{x}_i^{t,r-1}) - \nabla f_i\left(\boldsymbol{x}_i^{t,r-1}\right)\right)\right\|^2$$

$$= \mathbb{E}\left[\left\|\boldsymbol{x}_i^{t,r-1} - \boldsymbol{x}^t\right\|^2 - 2 < \boldsymbol{x}_i^{t,r-1} - \boldsymbol{x}^t, \eta_l \nabla F_i(\boldsymbol{x}_i^{t,r-1}) > + \left\|\eta_l \nabla F_i(\boldsymbol{x}_i^{t,r-1})\right\|^2\right] + \eta_l^2 \sigma_l^2$$

$$= \mathbb{E}\left[\left\|\boldsymbol{x}_i^{t,r-1} - \boldsymbol{x}^t\right\|^2 - 2 < \frac{1}{\sqrt{2R-1}}(\boldsymbol{x}_i^{t,r-1} - \boldsymbol{x}^t), \sqrt{2R-1}\eta_l \nabla F_i(\boldsymbol{x}_i^{t,r-1}) > + \left\|\eta_l \nabla F_i(\boldsymbol{x}_i^{t,r-1})\right\|^2\right] + \eta_l^2 \sigma_l^2$$

$$\leq \left(1 + \frac{1}{2R-1}\right)\mathbb{E}\left[\left\|\boldsymbol{x}_i^{t,r-1} - \boldsymbol{x}^t\right\|^2\right] + 2R\mathbb{E}\left[\left\|\eta_l \nabla F_i(\boldsymbol{x}_i^{t,r-1})\right\|^2\right] + \eta_l^2 \sigma_l^2$$

$$= \left(1 + \frac{1}{2R-1}\right)\mathbb{E}\left[\left\|\boldsymbol{x}_i^{t,r-1} - \boldsymbol{x}^t\right\|^2\right] + 2R\mathbb{E}\left[\left\|\eta_l\left(\nabla F_i(\boldsymbol{x}_i^{t,r-1}) \pm \nabla f\left(\boldsymbol{x}^t\right) \pm \nabla f_i\left(\boldsymbol{x}^t\right)\right)\right\|^2\right] + \eta_l^2 \sigma_l^2$$

$$\leq \left(1 + \frac{1}{2R-1}\right)\mathbb{E}\left\|\boldsymbol{x}_i^{t,r-1} - \boldsymbol{x}^t\right\|^2 + 6R\mathbb{E}\left[\left\|\eta_l\left(\nabla f_i\left(\boldsymbol{x}_i^{t,r-1}\right) - \nabla f_i\left(\boldsymbol{x}^t\right)\right)\right\|^2\right] + 6R\mathbb{E}\left[\left\|\eta_l\left(\nabla f_i\left(\boldsymbol{x}^t\right)\right)\right\|^2\right] + \eta_l^2 \sigma_l^2$$

$$\leq \left(1 + \frac{1}{2R-1} + 6R\eta_l^2 L^2\right)\mathbb{E}\left\|\boldsymbol{x}_i^{t,r-1} - \boldsymbol{x}^t\right\|^2 + \eta_l^2(\sigma_l^2 + 6R\sigma_g^2 + 6R\mathbb{E}\left[\left\|\nabla f\left(\boldsymbol{x}^t\right)\right\|^2\right])$$

Unrolling the recursion, we obtain

$$\mathbb{E}\left\|\boldsymbol{x}_i^{t,r} - \boldsymbol{x}^t\right\|^2 \leq \sum_{p=0}^{r-1}\left(1 + \frac{1}{2R-1} + 4R\eta_l^2 L^2\right)^p \eta_l^2(\sigma_l^2 + 6R\sigma_g^2 + 6R\mathbb{E}\left[\left\|\nabla f\left(\boldsymbol{x}^t\right)\right\|^2\right])$$

$$\leq (R-1)\left[\left(1 + \frac{1}{R-1}\right)^R - 1\right]\eta_l^2(\sigma_l^2 + 6R\sigma_g^2 + 6R\mathbb{E}\left[\left\|\nabla f\left(\boldsymbol{x}^t\right)\right\|^2\right]) \tag{29}$$

$$\leq 5R\eta_l^2(\sigma_l^2 + 6R\sigma_g^2 + 6R\mathbb{E}\left[\left\|\nabla f\left(\boldsymbol{x}^t\right)\right\|^2\right])$$

where we use the fact that $(1 + \frac{1}{R-1})^R \leq 5$ for $R > 1$. Then, we have

$$\sum_{i=1}^N \boldsymbol{\lambda}_i^2 \mathbb{E}\left[\left\|\boldsymbol{g}_i^t\right\|^2\right] \leq W \sum_{i=1}^N \boldsymbol{\lambda}_i \mathbb{E}\left[\left\|\boldsymbol{g}_i^t\right\|^2\right] \leq 5WR\eta_l^2(\sigma_l^2 + 6R\sigma_g^2 + 6R\mathbb{E}\left[\left\|\nabla f\left(\boldsymbol{x}^t\right)\right\|^2\right]) \tag{30}$$

where we use $W = \max\{\boldsymbol{\lambda}_i\}_{i\in[N]}$ and the fact by Assumption 4.3 that $\sum_{i=1}^N \boldsymbol{\lambda}_i \|\nabla F_i(\boldsymbol{x}^t)\|^2 \leq \|\nabla f(\boldsymbol{x}^t)\|^2 + \sigma_g^2$.

**Notation**. For simple notation, we use $\bar{\Delta} = 5R\eta_l^2(\sigma_l^2 + 6R\sigma_g^2 + 6R\mathbb{E}\left[\|f(\boldsymbol{x}^t)\|^2\right])$ to denote the upper bound of local drift.

$\square$

## C.2 Non-convex Analyses

Now we are ready to give our convergence analysis in detail.

*Proof.* We recall the updating rule during round $t$ as:

$$\boldsymbol{x}^{t+1} = \boldsymbol{x}^t - \eta_g \sum_{i\in S^t} \frac{\boldsymbol{\lambda}_i \boldsymbol{g}_i^t}{\boldsymbol{p}_i^t} = \boldsymbol{x}^t - \eta_g \boldsymbol{d}^t, \text{where } \boldsymbol{g}_i^t = \boldsymbol{x}^t - \boldsymbol{x}_i^{t,R} = \eta_l \sum_{r=1}^R \nabla F_i(\boldsymbol{x}_i^{t,r-1}).$$

**Notation.** For clear notation, we denote $W = \max\{\boldsymbol{\lambda}_i\}_{i\in[N]}$, $\tau_*^t = \frac{(N-K)\alpha_*^t + K}{K}$.

**Descent lemma.** Using the smoothness of $f$ and taking expectations conditioned on $x$ and over the sampling $S^t$, we have

$$\mathbb{E}\left[f(\boldsymbol{x}^{t+1})\right] = \mathbb{E}\left[f(\boldsymbol{x}^t - \eta_g \boldsymbol{d}^t)\right] \le \mathbb{E}[f(\boldsymbol{x}^t)] - \eta_g \mathbb{E}[\langle \nabla f(\boldsymbol{x}^t), \boldsymbol{d}^t \rangle] + \frac{L}{2}\eta_g^2 \mathbb{E}\left[\|\boldsymbol{d}^t\|^2\right]$$

$$= \mathbb{E}\left[f(\boldsymbol{x}^t - \eta_g \boldsymbol{d}^t)\right] \le \mathbb{E}[f(\boldsymbol{x}^t)] - \eta_g \mathbb{E}[\langle \nabla f(\boldsymbol{x}^t), \sum_{i=1}^{N} \boldsymbol{\lambda}_i \boldsymbol{g}_i^t \rangle] + \frac{L}{2}\eta_g^2 \mathbb{E}\left[\|\boldsymbol{d}^t\|^2\right]$$

$$\le \mathbb{E}[f(\boldsymbol{x}^t)] - \eta_g \mathbb{E}\|\nabla f(\boldsymbol{x}^t)\|^2 + \eta_g \mathbb{E}[\langle \nabla f(\boldsymbol{x}^t), \nabla f(\boldsymbol{x}^t) - \sum_{i=1}^{N} \boldsymbol{\lambda}_i \boldsymbol{g}_i^t \rangle] + \frac{L}{2}\eta_g^2 \mathbb{E}\left[\|\boldsymbol{d}^t\|^2\right] \qquad (31)$$

$$\le f(\boldsymbol{x}^t) - \frac{\eta_g}{2}\|\nabla f(\boldsymbol{x}^t)\|^2 + \underbrace{\frac{\eta_g}{2}\mathbb{E}\left[\|\nabla f(\boldsymbol{x}^t) - \sum_{i=1}^{N} \boldsymbol{\lambda}_i \boldsymbol{g}_i^t\|^2\right]}_{T_1} + \underbrace{\frac{L}{2}\eta_g^2 \mathbb{E}\left[\|\boldsymbol{d}^t\|^2\right]}_{T_2},$$

where the last inequality follows since $\langle a, b \rangle \le \frac{1}{2}\|a\|^2 + \frac{1}{2}\|b\|^2, \forall a, b \in \mathbb{R}^d$.

**Bounding $T_1$.** We first investigate the expectation gap between global first-order gradient and utilized global estimates,

$$\mathbb{E}\left[\left\|\sum_{i=1}^{N} \boldsymbol{\lambda}_i \nabla f_i(\boldsymbol{x}^t) - \sum_{i=1}^{N} \boldsymbol{\lambda}_i \boldsymbol{g}_i^t\right\|^2\right] = \mathbb{E}\left[\left\|\sum_{i=1}^{N} \boldsymbol{\lambda}_i \left(\nabla f_i(\boldsymbol{x}^t) - \boldsymbol{g}_i^t\right)\right\|^2\right]$$

$$= \mathbb{E}\left[\left\|\sum_{i=1}^{N} \boldsymbol{\lambda}_i \left(\nabla f_i(\boldsymbol{x}^t) - \eta_l \sum_{r=1}^{R} \nabla F_i(\boldsymbol{x}_i^{t,r-1})\right)\right\|^2\right]$$

$$= \mathbb{E}\left[\left\|\sum_{i=1}^{N} \boldsymbol{\lambda}_i \sum_{r=1}^{R} \left(\frac{1}{R}\nabla f_i(\boldsymbol{x}^t) - \eta_l \nabla F_i(\boldsymbol{x}_i^{t,r-1}) + \eta_l \nabla f_i(\boldsymbol{x}_i^{t,r-1}) - \eta_l \nabla f_i(\boldsymbol{x}_i^{t,r-1})\right)\right\|^2\right]$$

$$\le 2\mathbb{E}\left[\left\|\sum_{i=1}^{N} \boldsymbol{\lambda}_i \sum_{r=1}^{R} \left(\frac{1}{R}\nabla f_i(\boldsymbol{x}^t) - \eta_l \nabla f_i(\boldsymbol{x}_i^{t,r-1})\right)\right\|^2\right] \qquad (32)$$

$$+ 2\mathbb{E}\left[\left\|\sum_{i=1}^{N} \boldsymbol{\lambda}_i \eta_l \sum_{r=1}^{R} \left(\nabla f_i(\boldsymbol{x}_i^{t,r-1}) - \nabla F_i(\boldsymbol{x}_i^{t,r-1})\right)\right\|^2\right]$$

$$\le 2\mathbb{E}\left[\left\|\sum_{i=1}^{N} \boldsymbol{\lambda}_i \sum_{r=1}^{R} \left(\frac{1}{R}\nabla f_i(\boldsymbol{x}^t) - \eta_l \nabla f_i(\boldsymbol{x}_i^{t,r-1})\right)\right\|^2\right] + 2\eta_l^2 R \sigma_l^2$$

$$\le \frac{2N}{R^2}\sum_{i=1}^{N} \boldsymbol{\lambda}_i^2 \mathbb{E}\left[\left\|\sum_{r=1}^{R} \nabla f_i(\boldsymbol{x}^t) - \eta_l R \sum_{r=1}^{R} \nabla f_i(\boldsymbol{x}_i^{t,r-1})\right\|^2\right] + 2\eta_l^2 R \sigma_l^2$$

Then, we have

$$
\mathbb{E}\left[\left\|\sum_{i=1}^{N}\boldsymbol{\lambda}_i\nabla f_i(\boldsymbol{x}^t)-\sum_{i=1}^{N}\boldsymbol{\lambda}_i\boldsymbol{g}_i^t\right\|^2\right]
$$

$$
\leq \frac{2N}{R^2}\sum_{i=1}^{N}\boldsymbol{\lambda}_i^2\mathbb{E}\left[\left\|(1-\eta_l R)\sum_{r=1}^{R}\nabla f_i(\boldsymbol{x}^t)+\eta_l R\left(\sum_{r=1}^{R}\nabla f_i(\boldsymbol{x}^t)-\sum_{r=1}^{R}\nabla f_i(\boldsymbol{x}_i^{t,r-1})\right)\right\|^2\right]+2\eta_l^2 R\sigma_l^2
$$

$$
\leq \frac{4N}{R^2}\sum_{i=1}^{N}\boldsymbol{\lambda}_i^2\mathbb{E}\left[\left\|(1-\eta_l R)\sum_{r=1}^{R}\nabla f_i(\boldsymbol{x}^t)\right\|^2\right]
$$

$$
+\frac{4N}{R^2}\sum_{i=1}^{N}\boldsymbol{\lambda}_i^2\mathbb{E}\left[\left\|\eta_l R\left(\sum_{r=1}^{R}\nabla f_i(\boldsymbol{x}^t)-\sum_{r=1}^{R}\nabla f_i(\boldsymbol{x}_i^{t,r-1})\right)\right\|^2\right]+2\eta_l^2 R\sigma_l^2 \tag{33}
$$

$$
\leq 4N(1-\eta_l R)^2\sum_{i=1}^{N}\boldsymbol{\lambda}_i^2\mathbb{E}\left[\left\|\nabla f_i(\boldsymbol{x}^t)\right\|^2\right]+\frac{4N}{R^2}\eta_l^2\sum_{i=1}^{N}\boldsymbol{\lambda}_i^2 R\sum_{r=1}^{R}\mathbb{E}\left[\left\|\nabla f_i(\boldsymbol{x}^t)-\nabla f_i(\boldsymbol{x}_i^{t,r-1})\right\|^2\right]+2\eta_l^2 R\sigma_l^2
$$

$$
\leq 4N(1-\eta_l R)^2\sum_{i=1}^{N}\boldsymbol{\lambda}_i^2\mathbb{E}\left[\left\|\nabla f_i(\boldsymbol{x}^t)\right\|^2\right]+\frac{4N}{R^2}\eta_l^2 L^2\sum_{i=1}^{N}\boldsymbol{\lambda}_i^2 R\sum_{r=1}^{R}\mathbb{E}\left[\left\|\boldsymbol{x}^t-\boldsymbol{x}_i^{t,r-1}\right\|^2\right]+2\eta_l^2 R\sigma_l^2.
$$

$$
\leq 4N(1-\eta_l R)^2\sum_{i=1}^{N}\boldsymbol{\lambda}_i^2\mathbb{E}\left[\left\|\nabla f_i(\boldsymbol{x}^t)\right\|^2\right]+4N\eta_l^2 L^2\sum_{i=1}^{N}\boldsymbol{\lambda}_i^2\mathbb{E}\left[\left\|\boldsymbol{x}^t-\boldsymbol{x}_i^{t,r-1}\right\|^2\right]+2\eta_l^2 R\sigma_l^2.
$$

$$
\leq 8N(1-\eta_l R)^2 W(\mathbb{E}\left[\|\nabla f(\boldsymbol{x}^t)\|^2\right]+\sigma_g^2)+4N\eta_l^2 L^2 W\bar{\Delta}+2\eta_l^2 R\sigma_l^2
$$

$$
\leq 8(1-\eta_l R)^2(\mathbb{E}\left[\|\nabla f(\boldsymbol{x}^t)\|^2\right]+\sigma_g^2)+4\eta_l^2 L^2\bar{\Delta}+2\eta_l^2 R\sigma_l^2,
$$

where we plug equation 30 at the last and use the fact that $W$ is proportional to $\frac{1}{N}$ (omit constant factor $NW$). Then, we have

$$
T_1 \leq 4\eta_g(1-\eta_l R)^2(\mathbb{E}\left[\|\nabla f(\boldsymbol{x}^t)\|^2\right]+\sigma_g^2)+2\eta_g\eta_l^2 L^2\bar{\Delta}+\eta_g\eta_l^2 R\sigma_l^2. \tag{34}
$$

**Bounding $T_2$.** Now, we focus on the quality of estimates,

$$
\mathbb{E}\left[\|\boldsymbol{d}^t\|^2\right] \leq \mathbb{E}\left[\left\|\sum_{i\in S^t}\frac{\boldsymbol{\lambda}_i\boldsymbol{g}_i^t}{\boldsymbol{p}_i^t}-\sum_{i=1}^{N}\boldsymbol{\lambda}_i\boldsymbol{g}_i^t\right\|^2+\left\|\sum_{i=1}^{N}\boldsymbol{\lambda}_i\boldsymbol{g}_i^t\right\|^2\right]
$$

$$
\leq \underbrace{\mathbb{E}\left[\left\|\sum_{i\in S^t}\frac{\boldsymbol{\lambda}_i\boldsymbol{g}_i^t}{\boldsymbol{p}_i^t}-\sum_{i\in S^*}\frac{\boldsymbol{\lambda}_i\boldsymbol{g}_i^t}{\boldsymbol{p}_i^*}\right\|^2\right]}_{Q(S^t)}+\underbrace{\mathbb{E}\left[\left\|\sum_{i\in S^*}\frac{\boldsymbol{\lambda}_i\boldsymbol{g}_i^t}{\boldsymbol{p}_i^*}-\sum_{i=1}^{N}\boldsymbol{\lambda}_i\boldsymbol{g}_i^t\right\|^2\right]+\mathbb{E}\left[\left\|\sum_{i=1}^{N}\boldsymbol{\lambda}_i\boldsymbol{g}_i^t\right\|^2\right]}_{(A)}. \tag{35}
$$

Here, the $Q(S^t)$ indicates the discrepancy between applied sampling and optimal sampling. The term $(A)$ indicates the intrinsic gap for the optimal sampling to approach its targets and the quality of the targets for

optimization. Using definition in equation 27 and equation 28, we have

$$(A) = \mathbb{E}\left[\left\|\sum_{i \in S^*} \frac{\boldsymbol{\lambda}_i \boldsymbol{g}_i^t}{\boldsymbol{p}_i^*} - \sum_{i=1}^{N} \boldsymbol{\lambda}_i \boldsymbol{g}_i^t\right\|^2\right] + \mathbb{E}\left[\left\|\sum_{i=1}^{N} \boldsymbol{\lambda}_i \boldsymbol{g}_i^t\right\|^2\right]$$

$$\leq \alpha_*^t \frac{(N-K)N}{(N-1)K} \mathbb{E}\left[\sum_{i=1}^{N} \boldsymbol{\lambda}_i^2 \left\|\boldsymbol{g}_i^t\right\|^2\right] + \mathbb{E}\left[\left\|\sum_{i=1}^{N} \boldsymbol{\lambda}_i \boldsymbol{g}_i^t\right\|^2\right]$$

$$\leq \alpha_*^t \frac{(N-K)N}{(N-1)K} \sum_{i=1}^{N} \boldsymbol{\lambda}_i^2 \mathbb{E}\left[\left\|\boldsymbol{g}_i^t\right\|^2\right] + N \sum_{i=1}^{N} \boldsymbol{\lambda}_i^2 \mathbb{E}\left[\left\|\boldsymbol{g}_i^t\right\|^2\right]$$

$$= \left(\alpha_*^t \frac{(N-K)N}{(N-1)K} + N\right) \sum_{i=1}^{N} \boldsymbol{\lambda}_i^2 \mathbb{E}\left[\left\|\boldsymbol{g}_i^t\right\|^2\right]$$

$$\leq \left(\frac{\alpha_*^t(N-K)+K}{K}\right) \frac{N}{N-1} \sum_{i=1}^{N} \boldsymbol{\lambda}_i^2 \mathbb{E}\left[\left\|\boldsymbol{g}_i^t\right\|^2\right]$$

$$= \tau_*^t \frac{N}{N-1} \sum_{i=1}^{N} \boldsymbol{\lambda}_i^2 \mathbb{E}\left[\left\|\boldsymbol{g}_i^t\right\|^2\right]$$

$$\leq \tau_*^t \frac{N}{N-1} W \bar{\Delta}$$

where $\tau_*^t := \frac{\alpha_*^t(N-K)+K}{K} \in [1, \frac{N}{K}]$ as we defined before. Then, we have

$$T_2 \leq \frac{L}{2}\eta_g^2 Q(S^t) + L\eta_g^2 \tau_*^t W \bar{\Delta} \tag{36}$$

where we let $\frac{N}{2(N-1)} \leq 1$ the last inequality for simplicity of notation.

**Putting together.** Substituting corresponding terms in equation 31 with equation 34 and equation 36 to finish the descent lemma, we have

$$\mathbb{E}\left[f(\boldsymbol{x}^{t+1})\right] \leq f(\boldsymbol{x}^t) - \frac{\eta_g}{2}\|\nabla f(\boldsymbol{x}^t)\|^2$$
$$+ 4\eta_g(1-\eta_l R)^2(\mathbb{E}\left[\|\nabla f(\boldsymbol{x}^t)\|^2\right] + \sigma_g^2) + 2\eta_g\eta_l^2 L^2 \bar{\Delta} + \eta_g\eta_l^2 R\sigma_l^2$$
$$+ \frac{L}{2}\eta_g^2 Q(S^t) + L\eta_g^2 \tau_*^t W \bar{\Delta}.$$

Then, substituting $\bar{\Delta}$ from Lemma C.1, we rearrange the terms to obtain

$$\mathbb{E}\left[f(\boldsymbol{x}^{t+1})\right] \leq f(\boldsymbol{x}^t) + \frac{L}{2}\eta_g^2 Q(S^t) - \frac{\eta_g}{2}\left(1 - 8(1-\eta_l R)^2 - 120R^2L^2\eta_l^4 - 60R^2L\tau_*^t W\eta_g\eta_l^2\right)\|\nabla f(\boldsymbol{x}^t)\|^2$$
$$+ \eta_g\left(4(1-\eta_l R)^2 + 60R^2\eta_l^4 L^2 + 30R^2\eta_l^2 L\tau_*^t W\eta_g\right)\sigma_g^2 \tag{37}$$
$$+ \eta_g\left(10RL^2\eta_l^4 + R\eta_l^2 + 5R\eta_l^2 L\tau_*^t W\eta_g\right)\sigma_l^2.$$

Taking a full expectation on both side and rearranging equation 37 and setting $\eta_g \leq \frac{1}{L}$ to adapt $L$, we obtain

$$\rho^t \mathbb{E}\|\nabla f(\boldsymbol{x}^t)\|^2 \leq \frac{2(\mathbb{E}[f(\boldsymbol{x}^t)] - \mathbb{E}[f(\boldsymbol{x}^{t+1})])}{\eta_g} + \beta^t \eta_g + \epsilon + Q(S^t), \tag{38}$$

where we have

$$\rho^t = \left(1 - 8(1-\eta_l R)^2 - 120R^2L^2\eta_l^2(\eta_l^2 + \tau_*^t W)\right),$$
$$\beta^t = 10\left(R\eta_l^2 L\sigma_l^2 + 6R^2\eta_l^2 L\sigma_g^2\right)\tau_*^t W,$$
$$\epsilon = 8\left((1-\eta_l R)^2 + 15R^2\eta_l^4 L^2\right)\sigma_g^2 + 2R\eta_l^2\left(10\eta_l^2 L^2 + 1\right)\sigma_l^2.$$

Then, taking averaging of both sides of equation 38 over from time 1 to $T$, we have

$$\frac{1}{T}\sum_{t=1}^{T}\rho^t\mathbb{E}\|\nabla f(\boldsymbol{x}^t)\|^2 \leq \frac{2(\mathbb{E}\left[f(\boldsymbol{x}^1)-f(\boldsymbol{x}^{T+1})\right])}{T\eta_g} + \bar{\beta}\eta_g + \epsilon + \sum_{t=1}^{T}\frac{Q(S^t)}{T},$$

where $\bar{\beta} = \frac{1}{T}\sum_{t=1}^{T}\beta^t$. Then, taking upper bound $\mathbb{E}\left[f(\boldsymbol{x}^1)-f(x^{+\infty})\right] \leq M$, $\hat{\rho} := \min\{\rho^t\}_{t=1}^{T}$, setting $\eta_g = \sqrt{\frac{2M}{T\bar{\beta}}}$ to minimize the upper bound, we have

$$\min_{t\in[T]}\mathbb{E}\|\nabla f(\boldsymbol{x}^t)\|^2 \leq \frac{1}{T}\sum_{t=1}^{T}\frac{\rho^t}{\hat{\rho}}\mathbb{E}\|\nabla f(\boldsymbol{x}^t)\|^2 \leq \sqrt{\frac{8M\bar{\beta}+T\epsilon^2}{T\hat{\rho}^2}} + \frac{\frac{1}{T}\sum_{t=1}^{T}Q(S^t)}{\hat{\rho}},$$

which concludes the proof. $\qquad\square$

## D  Detail Proofs of Online Convex Optimization

### D.1  Vanising Hindsight Gap: Proof of Lemma 5.1

*Proof.* We first arrange the term (B) in Equation equation 9 as follows,

$$\min_{\boldsymbol{p}}\sum_{t=1}^{T}\ell_t(\boldsymbol{p}) - \sum_{t=1}^{T}\min_{\boldsymbol{p}}\ell_t(\boldsymbol{p}) = \min_{\boldsymbol{p}}\sum_{t=1}^{T}\sum_{i=1}^{N}\frac{\pi_t^2(i)}{\boldsymbol{p}_i} - \sum_{t=1}^{T}\min_{\boldsymbol{p}}\sum_{i=1}^{N}\frac{\pi_t^2(i)}{\boldsymbol{p}_i}. \tag{39}$$

Here, we recall our mild Assumption 5.1,

$$\pi_*(i) := \lim_{t\to\infty}\pi_t(i), \ \Pi_* := \sum_{i=1}^{N}\pi_*(i), \ \forall i \in [N].$$

Then, denoting $V_T(i) := \sum_{t=1}^{T}(\pi_t(i)-\pi_*(i))^2$, we bound the cumulative variance over time $T$ per client $i \in [N]$,

$$\begin{aligned}
\pi_{1:T}^2(i) &= \sum_{t=1}^{T}(\pi_*(i) + (\pi_t(i)-\pi_*(i)))^2 \\
&\leq T\cdot\pi_*^2(i) + 2\pi_*(i)\sum_{t=1}^{T}|\pi_t(i)-\pi_*(i)| + \sum_{t=1}^{T}(\pi_t(i)-\pi_*(i))^2 \\
&\leq T\cdot\pi_*^2(i) + 2\pi_*(i)\sqrt{T\cdot V_T(i)} + V_T(i) \\
&= T\left(\pi_*(i) + \sqrt{\frac{V_T(i)}{T}}\right)^2.
\end{aligned} \tag{40}$$

Using the Lemma 2.2 and non-negativity of feedback we have,

$$\min_{\boldsymbol{p}}\sum_{i=1}^{N}\frac{\pi_t^2(i)}{\boldsymbol{p}_i} = \frac{(\sum_{i=1}^{N}\pi_t(i))^2}{K}. \tag{41}$$

We obtain the upper bound of the first term in Equation equation 39,

$$\begin{aligned}
\min_{\boldsymbol{p}}\sum_{t=1}^{T}\sum_{i=1}^{N}\frac{\pi_t^2(i)}{\boldsymbol{p}_i} &= \min_{\boldsymbol{p}}\sum_{i=1}^{N}\frac{\pi_{1:T}^2(i)}{\boldsymbol{p}_i} = \frac{\left(\sum_{i=1}^{N}\sqrt{\pi_{1:T}^2(i)}\right)^2}{K} \\
&\leq \frac{T}{K}\left(\sum_{i=1}^{N}\pi_*(i) + \sum_{i=1}^{N}\sqrt{\frac{V_T(i)}{T}}\right)^2 \\
&= \frac{T}{K}\left(\Pi_*^2 + 2\Pi_*\sum_{i=1}^{N}\sqrt{\frac{V_T(i)}{T}} + \left(\sum_{i=1}^{N}\sqrt{\frac{V_t(i)}{T}}\right)^2\right),
\end{aligned} \tag{42}$$

where we use Lemma 2.2 in the second line, and Equation equation 40 in the third line.

Then, we bound the second term in Equation equation 39:

$$
\begin{aligned}
\Pi_*^2 = \sum_{i=1}^{N} \pi_*^2(i) &\leq \left( \frac{1}{T} \sum_{t=1}^{T} \sum_{i=1}^{N} \pi_t(i) \right)^2 \leq \frac{1}{T} \sum_{t=1}^{T} (\sum_{i=1}^{N} \pi_t(i))^2 \\
&= \frac{K}{T} \sum_{t=1}^{T} \min_{\boldsymbol{p}} \sum_{i=1}^{N} \frac{\pi_t^2(i)}{\boldsymbol{p}_i},
\end{aligned}
\tag{43}
$$

where the first inequality uses the average assumption, the third inequality uses Jensen's inequality, and the last inequality uses Equation equation 41.

Overall, we combine the results in Equation equation 42 and equation 43, and conclude the proof:

$$
\min_{\boldsymbol{p}} \sum_{t=1}^{T} \ell_t(\boldsymbol{p}) - \sum_{t=1}^{T} \min_{\boldsymbol{p}} \ell_t(\boldsymbol{p}) \leq \frac{T}{K} \left( \sum_{i=1}^{N} \sqrt{\frac{V_T(i)}{T}} \right) \left( 2\Pi_* + \sum_{i=1}^{N} \sqrt{\frac{V_t(i)}{T}} \right).
\tag{44}
$$

$\square$

## D.2 Regret of Full Information

**Theorem D.1** (Static regret with full information). *Under Assumptions 5.1, sampling a batch of clients with an expected size of $K$, and setting $\gamma = G^2$, the FTRL scheme in equation 10 yields the following regret,*

$$
\sum_{t=1}^{T} \ell_t(\boldsymbol{p}^t) - \min_{\boldsymbol{p}} \sum_{t=1}^{T} \ell_t(\boldsymbol{p}) \leq \left( \frac{22NG}{\bar{z}} + \frac{2\sqrt{6}NG}{K} \right) \sum_{i=1}^{N} \sqrt{\pi_{1:T}^2(i)} + \frac{22NG^2}{\bar{z}},
\tag{45}
$$

*where we note the cumulative feedback $\sqrt{\pi_{1:T}^2(i)} \leq \mathcal{O}(\sqrt{T})$ following Assumption 5.1.*

*Proof.* We considering a restricted probability space $\Delta = \{\boldsymbol{p} \in \mathbb{R}^N | \boldsymbol{p}_i \geq p_{\min}, \sum_{i=1}^{N} \boldsymbol{p}_i = K, \forall i \in [N]\}$ where $p_{\min} \leq K/N$. Then, we decompose the regret,

$$
\text{Regret}_{\text{FTRL}}(T) = \underbrace{\sum_{t=1}^{T} \ell_t(\boldsymbol{p}^t) - \min_{p \in \Delta} \sum_{t=1}^{T} \ell_t(\boldsymbol{p})}_{(A)} + \underbrace{\min_{p \in \Delta} \sum_{t=1}^{T} \ell_t(\boldsymbol{p}) - \min_{\boldsymbol{p}} \sum_{t=1}^{T} \ell_t(\boldsymbol{p})}_{(B)}.
\tag{46}
$$

We separately bound the above terms in this section. The bound of (A) is related to the stability of the online decision sequence by playing FTRL, which is given in Lemma D.1. Term (B) is bounded by the minimal results of directing calculation.

**Bounding (A)**. Without loss of generality, we introduce the stability of the online decision sequence from FTRL to variance function $\ell$ as shown in the following lemma(Kalai & Vempala, 2005) (similar proof can also be found in (Hazan, 2012; Shalev-Shwartz et al., 2012)).

**Lemma D.1.** *Let $\mathcal{K}$ be a convex set and $\mathcal{R} : \mathcal{K} \mapsto \mathbb{R}$ be a regularizer. Given a sequence of functions $\{\ell_t\}_{t \in [T]}$ defined over $\mathcal{K}$, then setting $\boldsymbol{p}^t = \arg\min_{\boldsymbol{p} \in \mathbb{R}^N} \sum_{\tau=1}^{t-1} \ell_\tau(\boldsymbol{p}) + \mathcal{R}(\boldsymbol{p})$ ensures,*

$$
\sum_{t=1}^{T} \ell_t(\boldsymbol{p}^t) - \sum_{t=1}^{T} \ell_t(\boldsymbol{p}) \leq \sum_{t=1}^{T} (\ell_t(\boldsymbol{p}^t) - \ell_t(\boldsymbol{p}^{t+1})) + (\mathcal{R}(\boldsymbol{p}) - \mathcal{R}(\boldsymbol{p}^1)), \forall \boldsymbol{p} \in \mathcal{K}.
$$

We note that $\mathcal{R}(\boldsymbol{p}) = \sum_{i=1}^{N} \gamma/\boldsymbol{p}_i$ in our work. Furthermore, $\mathcal{R}(\boldsymbol{p})$ is non-negative and bounded by $N\gamma/p_{\min}$ with $p \in \Delta$. Thus, the above lemma incurs,

$$
\sum_{t=1}^{T} \ell_t(\boldsymbol{p}^t) - \sum_{t=1}^{T} \ell_t(\boldsymbol{p}) \leq \underbrace{\sum_{t=1}^{T} (\ell_t(\boldsymbol{p}^t) - \ell_t(\boldsymbol{p}^{t+1}))}_{\text{Bounded Below}} + \frac{N\gamma}{p_{\min}}.
\tag{47}
$$

To simply the following proof, we assume that $0 < \pi_1(t) \le \pi_2(t) \le \cdots \le \pi_N(t), t \in [T]$ to satisfies Lemma B.6 without the loss of generality. The stability relies on the evolution of cumulative feedback $\pi_{1:t}^2(i)$ and hence relies on the index in solution $l_1, l_2$ according to Lemma 2.2. Following the Lemma B.6, we have

$$
\boldsymbol{p}_i^t = \begin{cases} 1, & \text{if } i \ge l_2^t, \\ z_t \frac{\sqrt{\pi_{1:t-1}^2(i)+\gamma}}{c_t}, & \text{if } i \in (l_1^t, l_2^t), \\ p_{\min}, & \text{if } i \le l_1^t, \end{cases} \tag{48}
$$

where $z_t = K - N + l_2^t - l_1^t \cdot p_{\min} \le K$ and $c_t = \sum_{i \in (l_1^t, l_2^t)} \sqrt{\pi_{1:t}^2(i) + \gamma} \le \sum_{i=1}^N \sqrt{\pi_{1:t}^2(i) + \gamma}$ is the normalization factor . Then, we investigate the first term in the above inequality,

$$
\sum_{t=1}^T (\ell_t(\boldsymbol{p}^t) - \ell_t(\boldsymbol{p}^{t+1})) \le \sum_{t=1}^T \sum_{i=1}^N \pi_t^2(i) \cdot \left( \frac{1}{\boldsymbol{p}_i^t} - \frac{1}{\boldsymbol{p}_i^{t+1}} \right).
$$

**Remark**. According to the above inequality, we note that the stability of online convex optimization is highly related to the changing probability. We can have a trivial upper bound $\sum_{t=1}^T (\ell_t(\boldsymbol{p}^t) - \ell_t(\boldsymbol{p}^{t+1})) \le \sum_{t=1}^T \sum_{i=1}^N \pi_t^2(i) \cdot (1/p_{\min} - 1)$, which indicates that the stability is restricted by $p_{\min}$. Solving the sampling probability requires sorting cumulative feedbacks $\pi_{1:t}^2(i)$, the combinations of client-index and $\boldsymbol{p}_i^t$ are dynamic. Hence, directly bounding the above equation generally can be difficult. To obtain a tighter bound for FTRL, we investigate the possible

**Lemma D.2.** *Assuming that $\boldsymbol{p}_i^t < \boldsymbol{p}_i^{t+1}$, for all $i \in [N], t \in [T-1]$, the upper bound of $\left( \frac{1}{\boldsymbol{p}_i^t} - \frac{1}{\boldsymbol{p}_i^{t+1}} \right)$ is given by:*

$$
0 \le \left( \frac{1}{\boldsymbol{p}_i^t} - \frac{1}{\boldsymbol{p}_i^{t+1}} \right) \le \frac{1}{\min(z_t, z_{t+1})} \left( \frac{c_t}{\sqrt{\pi_{1:t-1}^2(i) + \gamma}} - \frac{c_{t+1}}{\sqrt{\pi_{1:t}^2(i) + \gamma}} \right). \tag{49}
$$

*Proof.* For all $t \in [T]$, we have cumulative feedbacks $\pi_{1:t-1}(i)$, $i \in [N]$ on the server. The server is able to compute results equation 11. As we are interested in the upper bound, we assume $\boldsymbol{p}_i^t < \boldsymbol{p}_i^{t+1}$ and discuss the cases below:

- **Case 1**: letting $(\boldsymbol{p}_i^t, \boldsymbol{p}_i^{t+1}) = (p_{\min}, z_{t+1} \frac{\sqrt{\pi_{1:t}^2(i)+\gamma}}{c_{t+1}})$, we have

$$
\begin{aligned}
\frac{1}{\boldsymbol{p}_i^t} - \frac{1}{\boldsymbol{p}_i^{t+1}} &= \frac{1}{p_{\min}} - \frac{c_{t+1}}{z_{t+1}\sqrt{\pi_{1:t}^2(i)+\gamma}} \\
&\le \frac{c_t}{z_t\sqrt{\pi_{1:t-1}^2(i)+\gamma}} - \frac{c_{t+1}}{z_{t+1}\sqrt{\pi_{1:t}^2(i)+\gamma}}, \\
&\le \frac{1}{\min(z_t, z_{t+1})} \left( \frac{c_t}{\sqrt{\pi_{1:t-1}^2(i)+\gamma}} - \frac{c_{t+1}}{\sqrt{\pi_{1:t}^2(i)+\gamma}} \right),
\end{aligned}
$$

where the second inequality uses equation 24 indicating $p_{\min} \ge z_t \frac{\sqrt{\pi_{1:t-1}^2(i)+\gamma}}{c_t}$.

- **Case 2**: letting $(\boldsymbol{p}_i^t, \boldsymbol{p}_i^{t+1}) = (z_t \frac{\sqrt{\pi_{1:t-1}^2(i)+\gamma}}{c_t}, z_{t+1} \frac{\sqrt{\pi_{1:t}^2(i)+\gamma}}{c_{t+1}})$, equation 49 naturally holds.

- **Case 3**: letting $(\boldsymbol{p}_i^t, \boldsymbol{p}_i^{t+1}) = (z_t \frac{\sqrt{\pi_{1:t-1}^2(i)+\gamma}}{c_t}, 1)$, we can know that $1 \le z_{t+1} \frac{\sqrt{\pi_{1:t}^2(i)+\gamma}}{c_{t+1}}$ by equation 24 and prove the conclusion analogous to case 1.

- **Case 4**: analogous to the case 1 and 3, letting $(\boldsymbol{p}_i^t, \boldsymbol{p}_i^{t+1}) = (p_{\min}, 1)$, equation 49 naturally holds.

Summarizing all cases to conclude the proof.                                                    □

Using Lemma D.2, we are ready to bound the stability of the online decision sequence:

$$\sum_{t=1}^{T}(\ell_t(\boldsymbol{p}^t) - \ell_t(\boldsymbol{p}^{t+1})) = \sum_{t=1}^{T}\sum_{i=1}^{N}\pi_t^2(i) \cdot \left(\frac{c_t}{z_t\sqrt{\pi_{1:t-1}^2(i)+\gamma}} - \frac{c_{t+1}}{z_{t+1}\sqrt{\pi_{1:t}^2(i)+\gamma}}\right)$$

$$\leq \sum_{t=1}^{T}\sum_{i=1}^{N}\frac{\pi_t^2(i)\cdot c_t}{\min(z_t,z_{t+1})} \cdot \left(\frac{1}{\sqrt{\pi_{1:t-1}^2(i)+\gamma}} - \frac{1}{\sqrt{\pi_{1:t}^2(i)+\gamma}}\right) \qquad \triangleright c_t \leq c_{t+1}$$

$$\leq \sum_{t=1}^{T}\sum_{i=1}^{N}\frac{\pi_t^2(i)\cdot \tilde{c}_t}{\min(z_t,z_{t+1})\sqrt{\pi_{1:t}^2(i)+\gamma}} \cdot \left(\sqrt{1+\frac{\pi_t^2(i)}{\pi_{1:t-1}^2(i)+\gamma}} - 1\right)$$

$$\leq \frac{\tilde{c}_T}{2}\sum_{t=1}^{T}\sum_{i=1}^{N}\frac{1}{\min(z_t,z_{t+1})}\frac{\pi_t(i)^4}{\sqrt{\pi_{1:t}^2(i)+\gamma}\cdot(\pi_{1:t-1}^2(i)+\gamma)}, \qquad \triangleright\sqrt{1+x}-1\leq\frac{x}{2}$$

where the third line uses definition $c_t \leq \tilde{c}_t = \sum_{i=1}^{N}\sqrt{\pi_{1:t}^2(i)+\gamma}$.

Letting $\gamma = G^2$, we have that $\pi_{1:t}^2(i) \leq \pi_{1:t-1}^2(i) + \gamma$ and $\sqrt{\pi_{1:t}^2(i)} \leq \sqrt{\pi_{1:t}^2(i)+\gamma}$. We define $\bar{z} = \min\{z_t\}_{t=1}^{T}$ and conclude the bound,

$$\sum_{t=1}^{T}(\ell_t(\boldsymbol{p}^t) - \ell_t(\boldsymbol{p}^{t+1})) \leq \frac{\tilde{c}_T}{2}\sum_{t=1}^{T}\sum_{i=1}^{N}\frac{\pi_t(i)^4}{(\pi_{1:t}^2(i))^{\frac{3}{2}}}$$

$$= G \cdot \frac{\tilde{c}_T}{2\bar{z}}\sum_{i=1}^{N}\sum_{t=1}^{T}\frac{(\pi_t(i)/G)^4}{((\pi_{1:t}(1)/G)^2)^{\frac{3}{2}}} \qquad \triangleright\text{Lemma B.1}$$

$$\leq \frac{22NG}{\bar{z}}\sum_{i=1}^{N}\sqrt{\pi_{1:T}^2(i)+G^2} \qquad \triangleright\text{Definition of } \tilde{c}_T$$

$$\leq \frac{22NG}{\bar{z}}\sum_{i=1}^{N}\left(\sqrt{\pi_{1:T}^2(i)}+G\right)$$

(50)

Finally, we can get the final bound of (A) by plugging equation 50 into equation 47 and summarizing as follows,

$$\sum_{t=1}^{T}\ell_t(\boldsymbol{p}^t) - \sum_{t=1}^{T}\ell_t(\boldsymbol{p}) \leq \frac{22NG}{\bar{z}}\sum_{i=1}^{N}\left(\sqrt{\pi_{1:T}^2(i)}+G\right) + \frac{NG^2}{p_{\min}}.$$

**Bounding (B)**. Using Corollaries B.1 and B.2, we bound the term (B) as follows,

$$\min_{p\in\Delta}\sum_{t=1}^{T}\ell_t(\boldsymbol{p}) - \min_{\boldsymbol{p}}\sum_{t=1}^{T}\ell_t(\boldsymbol{p})$$

$$\leq \frac{K(\sum_{i=1}^{N}\sqrt{\pi_{1:T}^2(i)})^2}{(K-Np_{\min})^2} - \frac{(\sum_{i=1}^{N}\sqrt{\pi_{1:T}^2(i)})^2}{K}$$

$$\leq \left(\frac{K}{(K-Np_{\min})^2} - \frac{1}{K}\right) \cdot \left(\sum_{i=1}^{N}\sqrt{\pi_{1:T}^2(i)}\right)^2$$

(51)

$$\leq \frac{6Np_{\min}}{K^2} \cdot \left(\sum_{i=1}^{N}\sqrt{\pi_{1:T}^2(i)}\right)^2$$

In the last line, we use the fact that $\frac{1}{(1-x)^2} - 1 \leq 6x$ for $x \in [0,1/2]$. Hence, we scale the coefficient

$$\frac{K}{(K-Np_{\min})^2} - \frac{1}{K} = \frac{1}{K}\left[\frac{1}{(1-Np_{\min}/K)^2} - 1\right] \leq \frac{6Np_{\min}}{K^2},$$

where we let $p_{\min} \leq K/(2N)$.

**Summary**. Setting $\gamma = G^2$, and combining the bound in equation 47 and equation 51, we have,

$$\text{Regret}_{\text{FTRL}}(T) = \sum_{t=1}^{T} \ell_t(\boldsymbol{p}^t) - \min_{\boldsymbol{p}} \sum_{t=1}^{T} \ell_t(\boldsymbol{p})$$

$$\leq \frac{22NG}{\bar{z}} \sum_{i=1}^{N} \left( \sqrt{\pi_{1:T}^2(i)} + G \right) + \frac{NG^2}{p_{\min}} + \frac{6Np_{\min}}{K^2} \cdot \left( \sum_{i=1}^{N} \sqrt{\pi_{1:T}^2(i)} \right)^2. \tag{52}$$

The $p_{\min}$ is only relevant for the theoretical analysis. Hence, the choice of it is arbitrary, and we can set it to $p_{\min} = \min \left\{ K/(2N), GK/(\sqrt{6} \sum_{i=1}^{N} \sqrt{\pi_{1:T}^2(i)}) \right\}$ which turns the upper bound to the minimal value. Hence, we yield the final bound of FTRL in the end,

$$\sum_{t=1}^{T} \ell_t(\boldsymbol{p}^t) - \min_{\boldsymbol{p}} \sum_{t=1}^{T} \ell_t(\boldsymbol{p}) \leq \left( \frac{22NG}{\bar{z}} + \frac{2\sqrt{6}NG}{K} \right) \sum_{i=1}^{N} \sqrt{\pi_{1:T}^2(i)} + \frac{22NG^2}{\bar{z}}. \tag{53}$$

$\square$

### D.3 Expected Regret of Partial Feedback: Proof of Theorem 5.2

*Proof.* Using the property of unbiasedness, we have

$$\min_{\boldsymbol{p}} \mathbb{E}[\sum_{t=1}^{T} \ell_t(\tilde{\boldsymbol{p}}^t) - \sum_{t=1}^{T} \ell_t(\boldsymbol{p})]$$

$$= \min_{\boldsymbol{p}} \mathbb{E}[\sum_{t=1}^{T} \tilde{\ell}_t(\tilde{\boldsymbol{p}}^t) - \sum_{t=1}^{T} \tilde{\ell}_t(\boldsymbol{p})] \tag{54}$$

$$= \underbrace{\mathbb{E}\Big[ \sum_{t=1}^{T} \tilde{\ell}_t(\tilde{\boldsymbol{p}}^t) - \sum_{t=1}^{T} \tilde{\ell}_t(\boldsymbol{p}^t) \Big]}_{(A)} + \underbrace{\min_{\boldsymbol{p}} \mathbb{E}\Big[ \sum_{t=1}^{T} \tilde{\ell}_t(\boldsymbol{p}^t) - \sum_{t=1}^{T} \ell_t(\boldsymbol{p}) \Big]}_{(B)}.$$

**Bounding (A)**. We recall that $\tilde{\boldsymbol{p}}_i^t \geq \frac{\theta K}{N}$ for all $t \in [T], i \in [N]$ due to the mixing. Therefore, $\boldsymbol{p}_i^t \geq K/N$ implies $\tilde{\boldsymbol{p}}_i^t \geq K/N$. Thus, we have

$$\frac{1}{\tilde{\boldsymbol{p}}_i^t} - \frac{1}{\boldsymbol{p}_i^t} = \theta \cdot \frac{\boldsymbol{p}_i^t - \frac{K}{N}}{\tilde{\boldsymbol{p}}_i^t \boldsymbol{p}_i^t} \leq \theta \cdot \frac{\boldsymbol{p}_i^t}{\tilde{\boldsymbol{p}}_i^t \boldsymbol{p}_i^t} = \frac{\theta}{\tilde{\boldsymbol{p}}_i^t} \leq \theta \cdot \frac{N}{K}.$$

Moreover, if $\boldsymbol{p}_i^t \leq K/N$, the above inequality still holds. We extend the (A) as follows,

$$(A) := \mathbb{E}\Big[ \sum_{t=1}^{T} \tilde{\ell}_t(\tilde{\boldsymbol{p}}^t) - \sum_{t=1}^{T} \tilde{\ell}_t(\boldsymbol{p}^t) \Big]$$

$$= \mathbb{E}\Big[ \sum_{t=1}^{T} \sum_{i=1}^{N} \tilde{\pi}_t^2(i) \Big( \frac{1}{\tilde{\boldsymbol{p}}_i^t} - \frac{1}{\boldsymbol{p}_i^t} \Big) \Big]$$

$$\leq \theta \cdot \frac{N}{K} \cdot \mathbb{E}\Big[ \sum_{t=1}^{T} \sum_{i=1}^{N} \tilde{\pi}_t^2(i) \Big]$$

$$\leq \frac{\theta G^2 N^2}{K} T,$$

where we use $\mathbb{E}[\tilde{\pi}_t^2(i)] = \pi_t^2(i) \leq G^2$.

**Bounding (B)**. We note that $\boldsymbol{p}^t$ is the decision sequence playing FTRL with the mixed cost functions. Thus, we combine the mixing bound of feedback (i.e., $\tilde{\pi}_t^2(i) \leq \frac{G^2 N}{\theta K}$) and Theorem D.1. Replacing $G^2$ with $G^2 \frac{N}{\theta K}$, we get

$$\sum_{t=1}^T \tilde{\ell}_t(\boldsymbol{p}^t) - \min_{\boldsymbol{p}} \sum_{t=1}^T \tilde{\ell}_t(\boldsymbol{p}) \leq \left( \frac{22 N^{\frac{3}{2}} G}{\bar{z}\sqrt{\theta K}} + \frac{2\sqrt{6} N^{\frac{3}{2}} G}{\sqrt{\theta K^3}} \right) \mathbb{E}\left[ \sum_{i=1}^N \sqrt{\tilde{\pi}_{1:T}^2(i)} \right] + \frac{22 G^2 N^2}{\bar{z}\theta K}. \tag{55}$$

**Summary**. Using Jensen's inequality, we have $\mathbb{E}\left[ \sum_{i=1}^N \sqrt{\tilde{\pi}_{1:T}^2(i)} \right] \leq \sum_{i=1}^N \sqrt{\mathbb{E}[\tilde{\pi}_{1:T}^2(i)]} = \sum_{i=1}^N \sqrt{\pi_{1:T}^2(i)}$. Finally, we can get the upper bound of the regret in partial-bandit feedback,

$$\begin{aligned} N^2 \cdot \min_{\boldsymbol{p}} \mathbb{E}[\sum_{t=1}^T \ell_t(\tilde{\boldsymbol{p}}^t) - \sum_{t=1}^T \ell_t(\boldsymbol{p})] &\leq \frac{\theta G^2}{K} T + \left( \frac{22 G}{\bar{z}\sqrt{\theta N K}} + \frac{2\sqrt{6} G}{\sqrt{\theta N K^3}} \right) \mathbb{E}\left[ \sum_{i=1}^N \sqrt{\tilde{\pi}_{1:T}^2(i)} \right] + \frac{22 G^2}{\bar{z}\theta K} \\ &\leq \frac{\theta G^2}{K} T + \left( \frac{22 N^{\frac{1}{2}} G^2}{\bar{z}\sqrt{\theta K}} + \frac{2\sqrt{6} N^{\frac{1}{2}} G^2}{\sqrt{\theta K^3}} \right) \sqrt{T} + \frac{22 G^2}{\bar{z}\theta K}, \end{aligned} \tag{56}$$

where the last line uses the bound $\sum_{i=1}^N \sqrt{\pi_{1:T}^2(i)} \leq N G \sqrt{T}$. Now, we can optimize the upper bound of regret in terms of $\theta$. Notably, $\theta$ is independent on $T$ and we set $\theta = (\frac{N}{TK})^{\frac{1}{3}}$ to get the minimized bound. Additionally, we are pursuing an expected regret, which is $\text{Regret}_{(S)}(T)$ in the original definition in equation 9. Using the unbiasedness of the mixed estimation and modified costs, we can obtain the final bound:

$$\begin{aligned} N^2 \cdot \mathbb{E}[\text{Regret}_{(S)}(T)] &= \mathbb{E}[\sum_{t=1}^T \ell_t(\tilde{\boldsymbol{p}}^t) - \min_{\boldsymbol{p}} \sum_{t=1}^T \ell_t(\boldsymbol{p})] \\ &= \mathbb{E}[\sum_{t=1}^T \ell_t(\tilde{\boldsymbol{p}}^t) - \min_{\boldsymbol{p}} \sum_{t=1}^T \tilde{\ell}_t(\boldsymbol{p})] + \mathbb{E}[\min_{\boldsymbol{p}} \sum_{t=1}^T \tilde{\ell}_t(\boldsymbol{p}) - \min_{\boldsymbol{p}} \sum_{t=1}^T \ell_t(\boldsymbol{p})] \\ &\leq \mathcal{O}\big(N^{\frac{1}{3}} T^{\frac{2}{3}} / K^{\frac{4}{3}}\big) + \mathbb{E}[\min_{\boldsymbol{p}} \sum_{t=1}^T \tilde{\ell}_t(\boldsymbol{p}) - \min_{\boldsymbol{p}} \sum_{t=1}^T \ell_t(\boldsymbol{p})] \\ &\leq \tilde{\mathcal{O}}\big(N^{\frac{1}{3}} T^{\frac{2}{3}} / K^{\frac{4}{3}}\big), \end{aligned}$$

where the last inequality uses Lemma 5.1, and the conclusion in Theorem 8 (Borsos et al., 2018). It proves the second term induces an additional log term to the final bound.

**Remark.** Baseline works have additional averaging coefficient $\frac{1}{N^2}$ in their final bound. This is because they consider the weights $\boldsymbol{\lambda} = 1/N$ in stochastic optimization, while we include the $\lambda$ for clients' weights in federated optimization. To align with them, we omit the coefficient of $N^2$ and report the final bound for $\mathbb{E}[\text{Regret}_{(S)}(T)]$, as $N^2$ can be absorbed by excluding the $\boldsymbol{\lambda}$ from client feedback function $\pi(\cdot)$. $\qquad\square$

# E   Further Discussions

## E.1   A Sketch of Proof with Client Stragglers

We note the possibility that some clients are unavailable to participants due to local failure or being busy in each round. To extend our analysis to the case, we assume there is a known distribution of client availability $\mathcal{A}$ such that a subset $\mathcal{A}^t \sim \mathcal{A}$ of clients are available at the $t$-th communication round. Let $\boldsymbol{q}_i = \text{Prob}(i \in \mathcal{A}^t)$ denote the probability that client $i$ is available at round $t$. Based on the setting, we update the definition of estimation $\boldsymbol{d}^t$:

$$\boldsymbol{d}^t := \sum_{i \in S^t \subseteq \mathcal{A}^t} \frac{\boldsymbol{\lambda}_i \boldsymbol{g}_i^t}{\boldsymbol{q}_i \boldsymbol{p}_i^t},$$

where $S^t \subseteq \mathcal{A}^t$ indicates that we can only sample from available set. Then, we apply the estimation to variance and obtain the following target:

$$\text{Regret}(T) = \sum_{t=1}^{T} \sum_{i=1}^{N} \frac{\pi_t^2(i)}{q_i p_i} - \sum_{t=1}^{T} \min_{p} \sum_{i=1}^{N} \frac{\pi_t^2(i)}{q_i p_i}.$$

Analogous to our analysis in Appendix D, we could obtain a similar bound of the above regret that considers the availability.

### E.2 Differences between biased client sampling methods

This section discusses the main differences between unbiased client sampling and biased client sampling methods. The proposed K-Vib sampler is an unbiased sampler for the first-order gradient of objective 1. Recent biased client sampling methods include Power-of-Choice (POC) (Cho et al., 2020b) and DivFL (Balakrishnan et al., 2022). Concretely, POC requires all clients to upload local empirical loss as prior knowledge and selects clients with the largest empirical loss. DivFL builds a submodular based on the latest gradient from clients and selects clients to approximate all client information. Therefore, these client sampling strategies build a biased gradient estimation that may deviate from a fixed global goal.

FL with biased client sampling methods, such as POC and DivFL, can be considered dynamic re-weighting algorithms adjusting $p_i$. Analogous to the equation 1, the basic objective of FL with biased client sampling methods can be defined as follows (Li et al., 2020b; Balakrishnan et al., 2022; Cho et al., 2020b):

$$\min_{x \in \mathcal{X}} f(x) := \sum_{i=1}^{N} p_i f_i(x) := \sum_{i=1}^{N} p_i \mathbb{E}_{\xi_i \sim \mathcal{D}_i}[F_i(x, \xi_i)], \tag{57}$$

where $p$ is the probability simplex, and $p_i$ is the probability of client $i$ being sampled. The gradient estimation is defined as $g^t = \frac{1}{K} \sum_{i \in S^t} g_i$ accordingly. The targets of biased FL client sampling are determined by the sampling probability $p$ as a replacement of $\lambda$ in the original FedAvg objective 1. Typically, the value of $p$ is usually dynamic and implicit.

### E.3 Theoretical Comparison with OSMD

The K-Vib sampler proposed in this paper is orthogonal with the recent work OSMD sampler  Zhao et al. (2021b)[2] in theoretical contribution. We justify our points below:

a) According to Equations (6) and (7) in OSMD, it proposes an online mirror descent procedure that optimizes the additional estimates to replace the mixing strategy in Vrb Borsos et al. (2018). The approach can be also utilized as an alternative method in equation 12.

b) The improvement of the K-Vib sampler is obtained from the modification of the sampling procedure. In contrast, the OSMD still follows the conventional random sampling procedure, as we discussed in Lemma 2.1. Hence, our theoretical findings of applying the independent sampling procedure in adaptive client sampling can be transferred to OSMD as well.

In short, the theoretical improvement of our work is different from the OSMD sampler. And, our insights about utilizing the independent sampling procedure can be used to improve the OSMD sampler. Meanwhile, the OSMD also suggests future work for the K-Vib sampler in optimizing the additional estimates procedure instead of mixing.

## F   Experiments Details

The experiment implementations are supported by *FedLab* framework (Zeng et al., 2023b). We provide the missing experimental details below:

---

[2]we refer to the latest version `https://arxiv.org/pdf/2112.14332.pdf`

**Hyper-parameters Setting**. For all samplers, there is an implicit value $G$ (Lipschitz gradient) related to the hyper-parameters. We set $G = 0.01$ for the Synthetic dataset task and $G = 0.1$ for FEMNIST tasks. We set $\eta = 0.4$ for Mabs (Salehi et al., 2017) as suggested by the original paper. Vrb Borsos et al. (2018) also utilize mixing strategy $\theta = (N/T)^{\frac{1}{3}}$ and regularization $\gamma = G^2 * N/\theta$. For the case that $N > T$ in FEMNIST tasks, we set $\theta = 0.3$ following the official source code[3]. For Avare El Hanchi & Stephens (2020), we set $p_{\min} = \frac{1}{5N}$, $C = \frac{1}{\frac{1}{N} - p_{\min}}$ and $\delta = 1$ for constant-stepsize as suggested in Appendix D of original paper. For the K-Vib sampler, we set $\theta = (\frac{N}{TK})^{1/3}$ and $\gamma = G^2 \frac{N}{K\theta}$. We also fix $\gamma$ and $\theta = 0.3$ for our sensitivity study in Figure 6.

**Baselines with budget $K$.** Our theoretical results in Theorem 5.2 and empirical results in Figure 5 reveal a key improvement of our work, that is, the linear speed up in online convex optimization. In contrast, we provide additional experiments with the different budget $K$ in Figure 7. Baseline methods do not preserve the improvement property respecting large budget $K$ in adaptive client sampling for variance reduction. Moreover, with the increasing communication budget $K$, the optimal sampling value is decreasing. As a result, the regret of baselines increases in Figure 7, indicating the discrepancy to the optimal is enlarged.

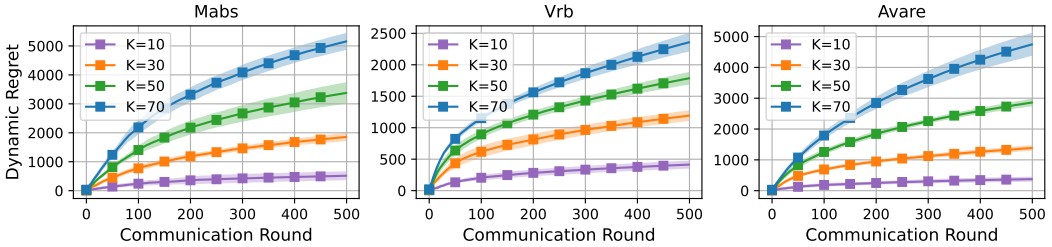

Figure 7: Regret of baseline algorithms with different $K$

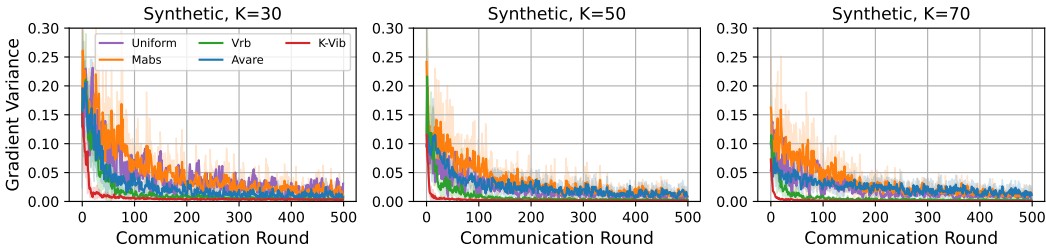

Figure 8: Gradient variance with different $K$

# G   Experiments on Text Datasets with Language Models

In this section, we evaluate the efficiency of samplers in practical experiments for training language models in the FL settings. We run all experiments over 5 random seeds.

**Language model pretraining on CCNews**[4]**.** Due to our computation resource limitation, we cannot conduct the original large-scale setting of benchmark (Charles et al., 2024). Analogously, we create a partitioned full CCNews of 708,241 training samples following benchmark (Charles et al., 2024) into $N = 1000$ clients. The data distribution across clients is shown in the left plot of Figure 9. Then, we train *from scratch* a GPT2 model (with the same architecture of Pythia-70M[5] (Biderman et al., 2023)) using the language modeling loss (i.e., next token prediction with cross-entropy). We set communication budget $K = 25$, local

---

[3]https://github.com/zalanborsos/online-variance-reduction

[4]https://huggingface.co/datasets/vblagoje/cc_news/tree/main

[5]https://huggingface.co/EleutherAI/pythia-70m

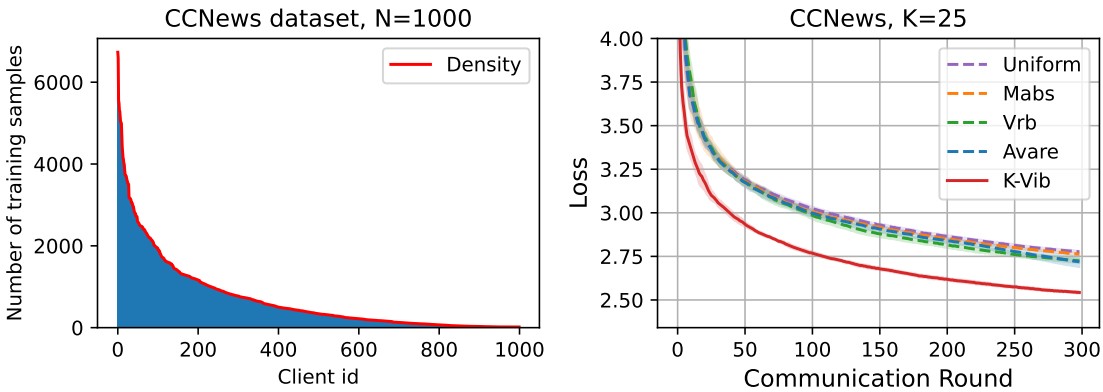

Figure 9: Experiments on CCNews dataset with scratch of Pythia-70M model.

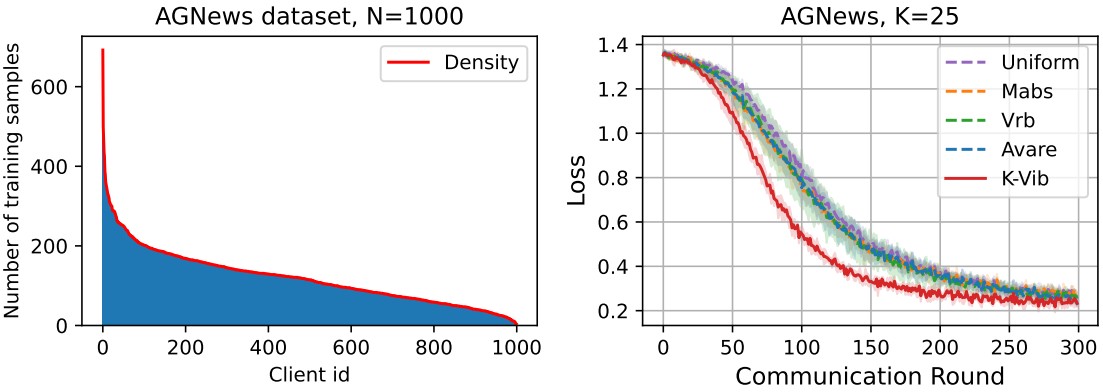

Figure 10: Experiments on AGNews dataset with DistillBert model.

learning rate $1e^{-4}$, batch size 16, and epoch 1 for the local SGD optimizer. We report the cross-entropy loss in the right plot of Figure 9.

**Language model finetuning on AGNews**[6]**.** We fine-tune a pretrained language model DistillBert[7] (Sanh et al., 2019) on a federated partitioned AGNews dataset (Zhang et al., 2015). The AGNews is a text classification task with 4 labels consisting of 119,999 train samples and 7,599 test samples. We partition the AGNews dataset into $N = 1,000$ clients and let their distribution across clients be heavy long tails as shown in Figure 10. We set communication budget $K = 25$, local learning rate $5e^{-4}$, batch size 16, and epoch 1 for the local SGD optimizer.

We observe that K-Vib achieves faster convergence than baseline methods, while baseline methods only implement a marginal improvement compared to uniform sampling. Our results prove that K-Vib can enhance real-world FL applications, especially language model pretraining.

# H    Efficient Implementation

In experiments, we do not find a heavy computation time increase compared to baselines as our experiments only involve thousands of clients. To guarantee practical usage for large-scale systems, we present efficient implementation details of K-Vib.

---

[6]https://huggingface.co/datasets/fancyzhx/ag_news
[7]https://huggingface.co/distilbert/distilbert-base-uncased

We can maintain a sorted list of cumulative local weights $[\omega(1), \omega(2), \ldots, \omega(N)]$ such that $\omega(i) \le \omega(j)$, $\forall i, j \in [N]$ in Algorithm 2. For each communication round, the server receives feedback values as a list $[\pi_t(j)], \forall j \in S^t$. Then, the server will traverse the feedback list. For each element in the list, the server conducts two main steps as below:

- Step 1: For each $j \in S^t$, server computes estimates $\tilde{\omega}(j) = \omega(j) + \pi_t^2(j)/p_j^t$. Then, the server uses *binary-search* to find the index $k$ such that $\omega(k) \le \tilde{\omega}(j) < \omega(k+1)$ in the cumulative local weights.

- Step 2: Then, server update $\omega(j) = \tilde{\omega}(j)$ and move the position of $\omega(j)$ behind $\omega(k)$ to update the weights sequence.

This implementation implements a time complexity of $\mathcal{O}(T \cdot K \cdot \log N)$, where $T$ is the communication round, $K$ is the communication budget, and $N$ is the number of clients. For each communication round $t \in [T]$, the server updates $K$ times of the list with each time cost $\mathcal{O}(\log N)$ to conduct one binary search.

