# OpenReview forum: "Enhanced Federated Optimization: Adaptive Unbiased Sampling with Reduced Variance"
_TMLR — Rejected by TMLR_

### Review · Reviewer_7LzV · 2024-05-10

**Summary Of Contributions:**

Existing federated learning methods leverage a random client sampling procedure (biased or unbiased). This paper argues that the existing random client sampling procedure is less efficient; they thus propose an independent sampling procedure. K-VIB is proposed as an adaptive sampler to reduce the cumulative variance of global estimates. The authors proved that K-VIB enjoys a near-linear speed-up over the existing bound. Empirical results also demonstrate the effectiveness of K-VIB.

**Audience:**

Yes

**Broader Impact Concerns:**

I don't seem to find any Broader Impact Concerns in the current draft.

**Claims And Evidence:**

Yes

**Requested Changes:**

The requested changes have been described in "Weaknesses". I will re-list them below:


- I would urge the authors to add larger-scale experiments, e.g., on ImageNet or on the datasets proposed in [1] (i.e., beyond computer vision/image classification tasks).
- It would also be important to report the wall clock time of the proposed K-VIB sampler.

**Strengths And Weaknesses:**

Strengths:
- This paper is well-written and well-motivated.
- The research direction of improving the client sampling strategies in federated learning is promising.
- Both theoretical analysis and empirical results are shown to demonstrate the performance of the proposed K-VIB sampler.

Weaknesses:
Overall I like this paper, the proposed technique seems to be solid. However, the scale and depth of the empirical results can be largely improved (which is my major concern, and I will elaborate it below).
- I understand that the paper's contribution is primarily theoretical. However, in 2024, presenting only MNIST scale experiments appears insufficient. To demonstrate that the method can impact real-world applications, more extensive experiments are necessary. The research community has made significant efforts to provide large-scale federated datasets [1]; therefore, why not utilize them?
- Only image classification tasks have been evaluated to demonstrate the effectiveness of the proposed K-VIB sampler.
- What will be the major wall clock time improvement of using the K-VIB sampler

[1] https://proceedings.neurips.cc/paper_files/paper/2023/file/662bb9c4dcc96aeaac8e7cd3fc6a0add-Paper-Datasets_and_Benchmarks.pdf

---

### Review · Reviewer_oAkY · 2024-05-30

**Summary Of Contributions:**

This submission introduces a new sampling estimator for federated learning from the theoretical perspective to reduce the variance and deviation of the gradient updates. To derive the estimator, the submission first proves the superiority of the independent sampling procedure (ISP) compared to the random sampling procedure (RSP) in terms of smaller variance. Then, given the communication budget, the submission defines the regret, proves convergence results with client samling, and then provides a regret bound. Experiments on a synthetic dataset and FEMNIST demonstrate the empirical effectiveness.

**Audience:**

Yes

**Broader Impact Concerns:**

No broader impact concerns.

**Claims And Evidence:**

Yes

**Requested Changes:**

It would be great to revise the manuscript to answer the questions and concerns in "Weaknesses" and "Minor".

**Strengths And Weaknesses:**

Strengths:
1. A comprehensive theoretical framework for analyzing and improving the variance in federated learning is presented.
2. A practical sampling mechanism (K-ViB sampler) is proposed and backed up with both rigorous theoretical and empirical evidence.
3. The submission is written with high quality and is generally easy to read.

Weaknesses:
1. The presentation could be further improved with more details provided in the main content, such as the abstraction of sampling mechanism with probability matrix (is it general to associate any sampling procedure with probability matrix and conduct the analysis?), the use of optimal factor (is optimal factor in use in the main content? why is the numerator squared and the denominator is not?), and the connection of $p_{min}$ and $\gamma$ in Eqn. (10) and Eqn. (11) (can we determine $p_{min}$ given $\gamma$ and how?).

2. The empirical experiments can be conducted at a larger scale. Currently, the experiments are limited to synthetic datasets and FEMNIST. It could be strengthened from experiments on ImageNet-scale datasets IMHO.

Minor:
On Page 5, line 2: "Then, the RSP outperforms ISP with a larger budget of $K=30$" I didn't observe so in Figure 1(a). Could you explain more?

Disclaimer: I am an outsider of FL and did not check the proofs. I may adjust my review and rating based on other reviewers' comments.

---

### Review · Reviewer_J4rQ · 2024-06-09

**Summary Of Contributions:**

In this work the author propose a new adaptive client sampler called K-Vib. The sampling probability is adaptively updated given the local client's feedback. Empirical results suggest that the proposed sampler is able to achieve much faster convergence compared to prior works on real world datasets.

**Audience:**

Yes

**Claims And Evidence:**

Yes

**Requested Changes:**

- Explain and clean up Assumption 5.1.
- Add discussion on additional hyperparameters.

**Strengths And Weaknesses:**

Strengths:
- The paper presents sufficient theoretical analysis to show how the proposed sampler improves upon arbitrary unbiased sampling rule.
- The empirical results seem to validate the improvement in convergence rate. The results on more realistic settings such as FEMNIST is also improssive.

Weaknesses:
- I found the paper a bit hard to read in some places. For example, Assumption 5.1 is quite complicated and takes sometime to parse.
- Assumption 5.1 seems quite restrictive. As $t\rightarrow\infty$, the gap between local and global feedback function seems to be 0. Does it still hold in the case where there's heterogeneity?
- The new sampler seems to come with two additional hyperparameters: $\gamma$ and $\theta$, which introduces additional complexity. Could the authors explain how they tune the hyperparameters in real world exps? (e.g. FEMNIST)

---

> ### Author Response · Authors · 2024-06-10
>
> Thanks for your valuable time and constructive comments. **We have revised our paper in response to the weakness.** Please see the modifications (highlighted in blue) about Assumption 5.1 and Section 7. Here are our summarized responses to the weaknesses:
>
> ### **Regarding weakness#1**
>
> Thank you very much for the feedback. We have carefully revised Assumption 5.1. We added more discussions and explanations of its implications.
>
> ### **Regarding weakness#2**
>
> This is a great question. Extreme heterogeneity may break the assumptions about some clients. The assumption expects that the local feedback sequence of each client will monotonically decrease in the applied federated learning procedures.  Fortunately, this assumption can be further justified by client clustering techniques or stable FedAvg variants as we discussed in the limitation part, Section 7.
>
> ### **Regarding weakness#3**
>
> We tune the hyperparameters following our theoretical analysis. We provide the discussion on additional hyperparameters in Section 7. More details about hyperparameters setting about FEMNIST experiments are given in Appendix F.
>
> We sincerely hope our responses have adequately addressed the weaknesses. We are open to further discussion and willing to make additional improvements if necessary.

---

### Decision · Action_Editor_Zi9Q · 2024-08-06

**Recommendation:** Reject

**Comment:**

This paper makes the following main contributions:
- advocates for the arbitrary sampling framework (Horváth & Richtárik, 2019), (Chen et al., 2020) for client sampling in federated learning,
- presents a convergence upper bound for FedAvg with arbitrary client sampling, showing the impact of the sampling quality measure on the convergence,
- extends the online learning VRB (Borsos et al., 2018) to multiple client sampling (K-ViB) and establishes a regret bound for this procedure,
- evaluates the K-ViB sampler in numerical experiments.

The majority of the reviewers recommend rejection of the manuscript in its current form.

However, I believe a revision could be considered if the authors either significantly enrich the empirical study or extend the theory to include end-to-end convergence guarantees (FedAvg + K-ViB), which would allow for a comparison of the convergence bounds with prior work.

As a minor comment:
It appears that Theorem 4.1 has a non-vanishing bias term under non-optimal sampling. This contrasts with other convergence results for FedAvg, which typically show convergence under uniform sampling.It would be beneficial if the authors could comment on the tightness of their bounds in the revision.

**Audience:**

The topic is clearly of interest to parts of the TMLR community.
In particular, the reviewers commended the promising research on improving client sampling strategies in federated learning.

However, the reviewers found the empirical results to be too preliminary to generate strong interest from the community. Additionally, the theoretical results in the various sections appear somewhat disjoint, which may limit the overall interest in the individual findings.

**Claims And Evidence:**

The reviewers found the presented results accurate and correct but not entirely convincing (the reviewers would have liked to see a bit more scale and depth in the empirical study).

**Resubmission Of Major Revision:**

The authors may consider submitting a major revision at a later time.